palaeontology/ecology/biogeography

reef corals, Last Interglacial, fossil bias, ecological niche modelling, climate change, conservation palaeobiology

**Author for correspondence:**
Lewis A. Jones
e-mail: l.jones16@imperial.ac.uk

# Coupling of palaeontological and neontological reef coral data improves forecasts of biodiversity responses under global climatic change

Lewis A. Jones[1], Philip D. Mannion[2], Alexander Farnsworth[3], Paul J. Valdes[3], Sarah-Jane Kelland[4] and Peter A. Allison[1]

[1]Department of Earth Science and Engineering, Imperial College London, South Kensington, London SW7 2AZ, UK
[2]Department of Earth Sciences, University College London, Gower Street, London WC1E 6BT, UK
[3]School of Geographical Sciences, University of Bristol, Bristol BS8 1TH, UK
[4]Getech Group Plc, Elmete Hall, Elmete Lane, Leeds LS8 2LJ, UK

LAJ, 0000-0003-3902-8986; PDM, 0000-0002-9361-6941; AF, 0000-0001-5585-5338; PJV, 0000-0002-1902-3283; PAA, 0000-0002-4997-5314

Reef corals are currently undergoing climatically driven poleward range expansions, with some evidence for equatorial range retractions. Predicting their response to future climate scenarios is critical to their conservation, but ecological models are based only on short-term observations. The fossil record provides the only empirical evidence for the long-term response of organisms under perturbed climate states. The palaeontological record from the Last Interglacial (LIG; 125 000 years ago), a time of global warming, suggests that reef corals experienced poleward range shifts and an equatorial decline relative to their modern distribution. However, this record is spatio-temporally biased, and existing methods cannot account for data absence. Here, we use ecological niche modelling to estimate reef corals' realized niche and LIG distribution, based on modern and fossil occurrences. We then make inferences about modelled habitability under two future climate change scenarios (RCP4.5 and RCP8.5). Reef coral ranges during the LIG were comparable to the present, with no prominent equatorial decrease in habitability. Reef corals are likely to experience poleward range expansion and large equatorial declines under RCP4.5 and RCP8.5. However, this range expansion is probably optimistic in the face of anthropogenic climate

change. Incorporation of fossil data in niche models improves forecasts of biodiversity responses under global climatic change.

## 1. Introduction

Anthropogenic climate change, with projected global warming of 2–4.8°C, is the principal challenge of the twenty-first century [1]. Rising sea-surface temperatures (SSTs), associated with global warming, have already generated ecological disturbances and substantial range shifts of marine organisms [2–4]. Today, reef corals house some of the most biologically diverse and economically valuable ecosystems in the marine realm. Thermal stress, as a direct result of global warming, has generated unprecedented mass bleaching of reef corals, with three pan-tropical events recorded in the last two decades, leading to a decline in abundance, diversity and habitat structure [5–7]. Global warming has also led to clear poleward range expansion of some corals [8–12], although the evidence for equatorial range retractions is debatable [9,13]. However, because studies on extant species are based on short-term observations, during which time a taxon might never have occupied its full geographical or environmental space, it is difficult to predict whether this expansion/retraction trend is a function of global warming, or simply reflects the ongoing response of a species to a pre-existing forcing already applied to the system. Greater understanding of the range dynamics of reef corals is, therefore, critical to their conservation.

Several studies have used ecological niche modelling (ENM) to evaluate the environmental controls on coral reefs and predict global distribution under future climate change scenarios [14–17]. However, the fossil record provides the only empirical evidence for the long-term responses of organisms to environmental perturbations (e.g. [18]), offering a ground truth for model projections under varying climate scenarios, and potentially enabling additional insights into niche hypervolume (the full multi-dimensional abiotic space which a taxon might occupy indefinitely), commonly referred to as the fundamental niche [19]. Furthermore, it has been demonstrated that the integration of both palaeontological and neontological data improves biodiversity risk assessments [20–23]. Future projections trained solely on modern occurrences might therefore underestimate habitability due to a restricted present-day range in climatic variation. Through incorporation of the fossil record, more of this variation is captured. Thus, palaeontological data from warmer intervals should improve projections to future climate scenarios that are closer analogues than the present day.

Apparent range shifts have been observed in the Quaternary (2.58–0 Ma) fossil record of reef corals [12,13,24,25], with one study [13] demonstrating poleward range shifts and an equatorial decline in reef corals during the Last Interglacial (LIG), *ca* 125 000 years ago. Several authors have provided approximations on global SSTs for the LIG, with estimates ranging from no significant difference to as much as 2°C warmer than today [26–28]. Kiessling *et al.* [13] proposed that this temperature rise was the driver of the dramatic range shifts observed in LIG reef corals. Consequently, these authors suggested that anthropogenic global warming today means that an equatorial decline of reef corals is likely to follow their current poleward expansion, which has important implications for conservation, such as which geographical regions should be prioritized. However, the fossil record is inherently biased both spatially and temporally, impacted by incomplete sampling, variable fossil preservation, erosion and burial. Although existing subsampling methods, such as those applied by Kiessling *et al.* [13], can ameliorate problems associated with uneven 'raw' occurrence data (e.g. [29]), they cannot account for the absence of data. ENM offers the prospect of mitigating for biases caused by the uneven distribution of known occurrences (i.e. raw data) by measuring habitat availability, providing insights into the potential geographical distribution of organisms through consideration of their observed realized niche.

Here, we use ENM to: (i) evaluate the extent to which the LIG equatorial decline of reef corals is the result of fossil bias or a genuine loss in habitability; (ii) quantify the variation in geographical range of reef corals during the LIG and under future climate scenarios, with implications for understanding the evolution of the reef coral latitudinal biodiversity gradient; and (iii) demonstrate the value of using fossil reef coral data to improve biodiversity risk assessment through consideration of the realized niche under varied climate states.

## 2. Material and methods

### 2.1. Biogeographic data

When implementing ENM, most studies have focused on analyses of individual species [20,30–33]. However, due to the relative rarity of fossil occurrence data at global scale, we use occurrence data at

ecotype level (i.e. a group which shares specific environmental preferences). The responses of modern coral reefs to environmental variables are broadly similar at a global level, recognized as being limited to warm, clear and shallow water habitats [34].

Modern coral occurrences were downloaded on 5 July 2017 from the Ocean Biogeographic Information System (OBIS; www.iobis.org) using the search criteria 'Scleractinia', yielding 589 817 occurrences. Fossil coral occurrence data were accessed on 19 July 2017 through the Paleobiology Database (PBDB; www.paleobiodb.org). The PBDB returned 8021 occurrences for Pleistocene Scleractinia. These datasets were then filtered to exclude data not identified to species level, as well as further quality control checks (e.g. synonymizations). We based our analyses on zooxanthellate corals, herein referred to as reef corals, which host unicellular organisms known as zooxanthellae in their tissue, forming a photosymbiotic relationship [35]. Assignment to zooxanthellate class focused on the principle of uniformitarianism for fossil coral taxa. Azooxanthellate (those without a photosymbiotic relationship) and apozooxanthellate (those which may or may not sustain a photosymbiotic relationship) corals were excluded from both databases. Further, quality control was performed on the OBIS dataset, whereby occurrences found to occur on land or at depths greater than 100 m were removed under the assumption that there were errors in their geographical coordinates, species identification or are in fact fossil occurrences. In addition, we omitted data collected from rapid ecological assessments to improve the credibility of species-level identification, as well as those from coral transplant studies. The PBDB data were filtered to exclude occurrences which cannot be reliably assigned to the LIG. Duplicate occurrence records of the same species from exactly the same coordinates were also removed from both datasets.

The resultant modern occurrence dataset contains 700 zooxanthellate species, comprising 48 577 occurrences with 3553 unique coordinates, while the LIG dataset contains 190 zooxanthellate species, comprising 1337 occurrences (from 301 collections) with 166 unique coordinates (figure 1; electronic supplementary material, SM1). In preparation of modelling, we clipped all occurrences by environmental layers to correspond with the environmental data. To account for species that have become extinct since the LIG, or that lack modern occurrence records in the OBIS, we excluded species which do not occur in both the clipped fossil and modern datasets. The final modern and fossil datasets comprise 157 shared species, composed of 22 726 modern reef coral occurrences and 1022 LIG reef coral fossil occurrences. Finally, point occurrences were spatially subsampled at a raster resolution of $1.25° \times 1.25°$ to account for sampling bias. Subsampling resulted in the retention of 295 presence sites for modern reef corals and 56 for LIG reef corals (note that this does not affect the overall distribution, but prevents multiple records being excessively weighted during model training) (figure 1).

## 2.2. Environmental data

Climate variables for the modern (pre-industrial), LIG and two future (2100 AD) climate scenario simulations were derived from HadCM3M2.1N [36], a version of UK Met Office HadCM3, a coupled atmosphere–ocean General Circulation Model (GCM). HadCM3 has been heavily used as part of the Intergovernmental Panel on Climate Change (IPCC) 4th and 5th assessment reports [37] and successfully applied to studies of both present-day and LIG climate and biodiversity [30,36,38–42]. Here, we use IPCC representative concentration pathways (RCP) 4.5 and 8.5 for our future climate scenario simulations [43]; a description of both is provided in electronic supplementary material, SM2. To further improve our ENMs, bathymetric data were extracted from Getech's present-day digital elevation model (DEM), which provides gridded representations of both the Earth's topography and bathymetry. A description of both the GCM and DEM used in this study is provided in electronic supplementary material, SM2, along with data preparation protocols.

Environmental variables known to be ecologically limiting for modern corals (i.e. SST, salinity, irradiance and bathymetry) [14,34,35], and that can be viably determined for the fossil record from GCM outputs, were chosen to reflect abiotic niche characteristics (electronic supplementary material, SM2: table SM2). Initially, these variables were considered at several temporal levels: annual, seasonal and monthly. However, to reduce co-linearity between variables and prevent over-fitting, we retained the combinations of climate variables with a Pearson's pairwise correlation coefficient of less than 0.7 (electronic supplementary material, SM2: figures SM2–SM7). After enquiry, five environmental variables were used for final niche analysis: (i) mean annual SST; (ii) mean December–February irradiance; (iii) mean June–August irradiance; (iv) mean annual salinity; and (v) bathymetry. While we acknowledge that additional variables, in particular, aragonite saturation state, might be valuable

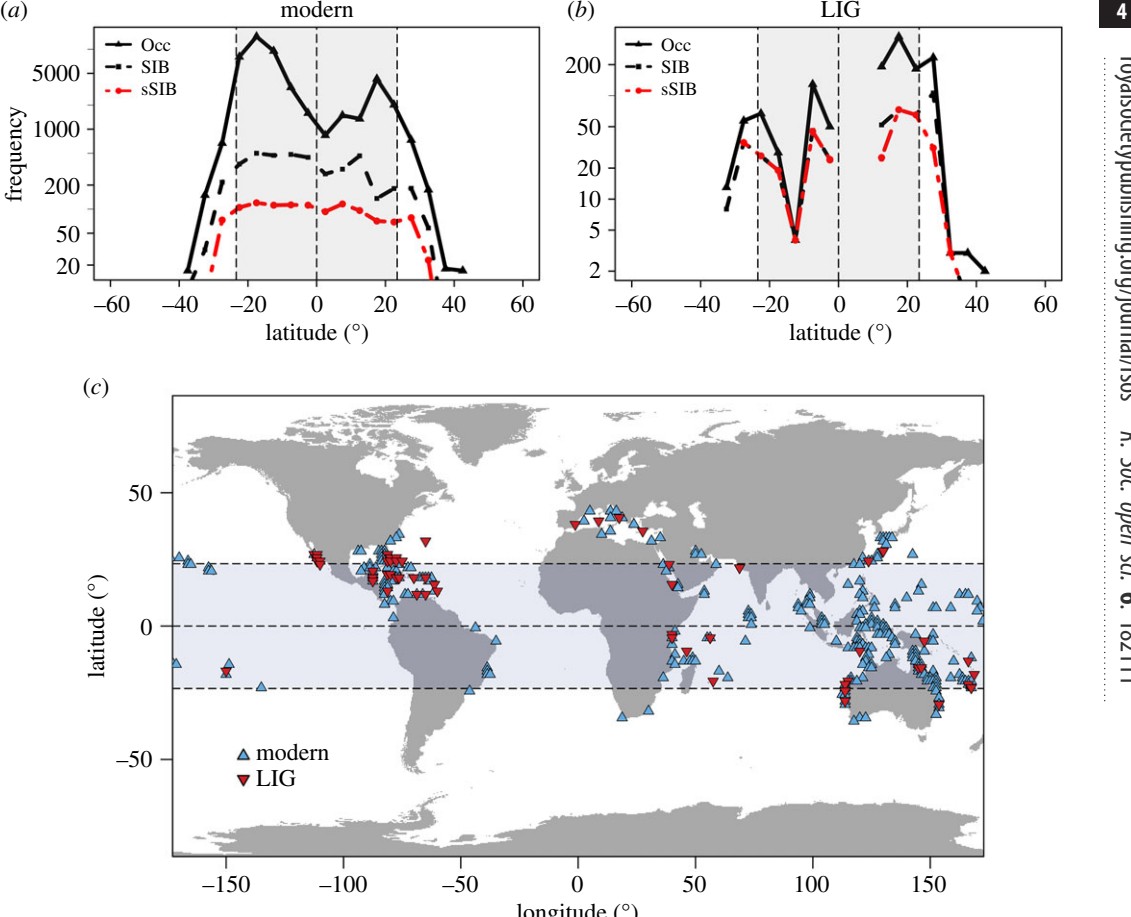

**Figure 1.** Latitudinal and global distribution of modern and LIG reef coral occurrences. (*a,b*) Frequency of occurrence records (Occ), species diversity (SIB) and shared species (species present in both time slices (sSIB)) of reef corals binned at 5° latitudinal zones. (*c*) World distribution map of shared species (subsampled at 1.25° × 1.25° and clipped to environmental data) between the modern and the LIG. Grey shaded area represents the extent of the tropics in the present day. Dashed lines indicate the Equator, the Northern Tropic and the Southern Tropic.

for estimating reef coral distribution, not all variables can be viably approximated for the geological past. Despite this, several biophysical studies have shown that the fundamental niche of an organism can be reasonably approximated by a limited set of biologically important variables (e.g. [44]). In addition, Couce *et al*. [14,15] demonstrated that high SST was the primary control on coral reef distribution, with high light availability, salinity, aragonite saturation state and low nutrient concentrations being of secondary significance. Nevertheless, we ran sensitivity analyses to compare modern projections from modern-trained models with and without aragonite saturation state to gauge the potential impact upon our results (provided in electronic supplementary material, SM2).

## 2.3. Niche modelling and quantifying range dynamics

### 2.3.1. Niche comparisons

As previously noted, short-term observations of distribution dynamics can be inherently biased by a taxon not occupying its full geographical or environmental space. However, an additional fundamental question is whether environmental niches are conserved over evolutionary time scales [45]. To evaluate this, we used the R-package ECOSPAT [46] to quantify measures of niche margin dynamics (unfilling, expansion and stability). We also performed niche overlap analyses, testing for niche equivalency and similarity of the reef coral niches occupied across two temporal windows (LIG and modern). The ECOSPAT niche overlap analysis applies kernel smoothers to the densities of

species occurrences in gridded environmental space, using a randomization test framework to compare observed equivalency and similarity to that anticipated under a null hypothesis [46]. Both niche equivalency and similarity are quantified in terms of Schoener's D [47], with values ranging from 0 (least equivalent or similar) to 1 (most equivalent or similar). We compared the subsampled occupied niches of LIG and modern reef corals using a principal component analysis (PCA-env), testing the null hypothesis by randomly sampling 5000 replicates of both LIG and modern niches. The PCA-env was calibrated with all five environmental variables implemented in the construction of our ENMs, and allowed us to assess differences in the occupied niche of reef corals from the LIG and modern (see [48] for detailed descriptions of the ECOSPAT package).

### 2.3.2. Ecological niche modelling

To map the potential distribution of reef corals in the modern, LIG and under future climate change scenarios (RCP4.5 and RCP8.5), we implemented three ENM methods: BIOCLIM [49], randomForest [50] and MaxEnt v. 3.4.1 [51]. In particular, MaxEnt has been shown to perform well in previous studies on coral reefs, as well as those using the fossil record for other taxonomic groups [14,16,30,32,52,53]. ENMs were generated using the R-package 'biomod2' v. 3.3-7 [54,55], running 100 subsample replicates with 75% of occurrences for model training, while 25% were reserved (randomly seeded) for model testing. We generated two model classes, training the ENM with both the LIG and modern occurrences, offering insights into the occupied niche and distribution under two diverse climate states. Using the three ENM methods, we produced an ensemble model based on mean projections from all modelling algorithms for each climate scenario, and for each model class (LIG- and modern-trained). As reef corals reside at the upper thermal limit of the oceans, we also built clamping masks for model projections to identify geographical areas which indicate environmental variable values outside of the training range.

### 2.3.3. Range comparisons

In order to measure distribution dynamics, we first generated 24 binary maps from the ensemble models using two different training datasets (LIG reef corals and modern reef corals) and three different binary thresholds for the four climate scenarios (LIG, modern, RCP4.5 and RCP8.5). We selected three thresholds for binary conversion: the 5th-, 10th- and 20th-percentiles of the lowest tail of habitat suitability values (5LPT, 10LPT, 20LPT) associated with the occurrence data for each model to provide insight into habitability dynamics (see electronic supplementary material, SM2, for further details). As short-term observations are intrinsically biased and perhaps incomplete in both environmental and geographical space, we combined binary predictions generated from both LIG and modern training datasets. This was done for each time-slice/climate scenario to establish a multi-temporal approach, estimating the full geographical localities reef corals may occupy. We then assessed whether this approach improved the models' capability to predict the occurrence of both LIG and modern localities, which would have implications for forecasting under future climate scenarios. The latitudinal range and habitability of reef corals for past, present and future climatic scenarios were computed as a function of the sum of suitable cells, within $5°$ latitudinal bins, for each model class' projections (LIG, modern and combined).

## 2.4. Model validation

Model performance was assessed using the area under the curve (AUC) statistic [56], derived from receiver operating characteristic (ROC) analyses, and the true skill statistic (TSS) [57]. In general, evaluation criteria for AUC values have been interpreted as excellent (0.90–1.00), very good (0.8–0.9), good (0.7–0.8), fair (0.6–0.7) and poor (0.5–0.6), whereas a score of less than 0.5 is worse than one would expect from a random model [58]. Evaluation scores for TSS range from −1 to 1, where a score of 1 demonstrates perfect model performance, while less than 0 is considered worse than a random model [57]. As we used two ENM constructs (LIG- and modern-trained), and generated respective forecasting and hindcasting (backtesting) projections, we calculated the AUC scores using LIG occurrences and modern occurrences as an evaluation dataset for model projections to other climate scenarios (i.e. modern-trained and LIG projection; LIG-trained and modern projection).

# 3. Results

## 3.1. Niche analyses

Niche margin analysis (electronic supplementary material, SM2: figure SM9) indicates considerable niche stability (91%) between LIG and modern reef corals, with minimal expansion (8%) and unfilling (5%). Observed niche overlap values for niche equivalency tests (Schoener's $D$) fall within the null distribution of the environment available to reef corals ($p > 0.05$), and thus we cannot reject the null hypothesis of niche equivalency. However, the observed overlap value for the niche similarity tests is greater than 95% ($p < 0.05$) of the simulated values, indicating modern reef corals occupy environments similar to those of the LIG.

## 3.2. Ecological niche modelling

### 3.2.1. Model performance

On the basis of the mean AUC scores (electronic supplementary material, SM2: figure SM11), the MaxEnt models performed best for both modern-trained (MaxEnt: 0.974, randomForest: 0.963, BIOCLIM: 0.843) and LIG-trained (MaxEnt: 0.976, randomForest: 0.950, BIOCLIM: 0.807) models, across individual projections. Conferring to AUC evaluation criteria [58], all models performed excellently with the exception of the BIOCLIM models, which can still be considered as very good (see electronic supplementary material, SM2: figures SM12 and SM13 for ROC plots). TSS scores reflect similar results in terms of model performance for modern-trained (MaxEnt: 0.899, RandomForest: 0.866, BIOCLIM: 0.685) and LIG-trained ENMs (MaxEnt: 0.924, RandomForest: 0.832, BIOCLIM: 0.615), with all models performing excellently, except BIOCLIM (electronic supplementary material, SM2: figure SM11). The modern-trained ensemble model had AUC scores of 0.979, 0.975 and 0.961 (electronic supplementary material, SM2: figure SM14) for the full occurrence dataset, test dataset and evaluation dataset, respectively. The LIG-trained model had AUC scores of 0.985, 0.984 and 0.965 (electronic supplementary material, SM2: figure SM14). In addition, combined modern binary projections were able to predict 95, 91 and 83% of modern occurrences for 5LPT, 10LPT and 20LPT, respectively, while LIG-trained-only projections were able to predict 62, 59 and 51%. Similarly, combined LIG binary projections were able to predict 98, 95 and 88% of LIG fossil occurrences for 5LPT, 10LPT and 20LPT, whereas modern-trained-only projections were able to predict 93, 79 and 54%.

### 3.2.2. Geographical predictions

The results from binary habitability maps (figure 2; raw, binary and clamping mask maps available in electronic supplementary material, SM2: figures SM15–22) for models trained on the modern occurrence dataset predict a larger habitable region at low latitudes (0–30°) than the LIG-trained ENM for all climate scenarios, converging only at higher latitudes. Models trained on the LIG occurrence dataset consistently calculate a smaller sum of habitable cells within the tropics than the modern-trained ENM. Combining model outputs (electronic supplementary material, SM2: figures SM23 and SM24) for both modern-trained and LIG-trained ENMs shows no distinctive increase in habitable cells at varying latitudes for all climate scenarios under the binary thresholds 5LPT and 10LPT. However, under the binary conversion threshold 20LPT (figure 3), the combined data increase the predicted habitability substantially at higher latitudes in the Northern Hemisphere, yet provide no such significant results for lower latitudes or in the Southern Hemisphere.

Threshold selection for binary conversion has important consequences on predicted habitability for future climate scenarios. When comparing predictions from modern-trained ENMs and different binary thresholds, LIG and modern habitability predictions reduce uniformly between thresholds, whereas RCP4.5 and RCP8.5 do not (figure 2). When applying binary threshold 5LPT, global habitability seems to increase under RCP4.5 and RCP8.5. However, under 20LPT, there is a substantial reduction in habitability relative to the modern and LIG. These results suggest that under RCP4.5 and RCP8.5, there will be an increase in marginal habitats for reef corals, and a prominent decrease in those more favourable.

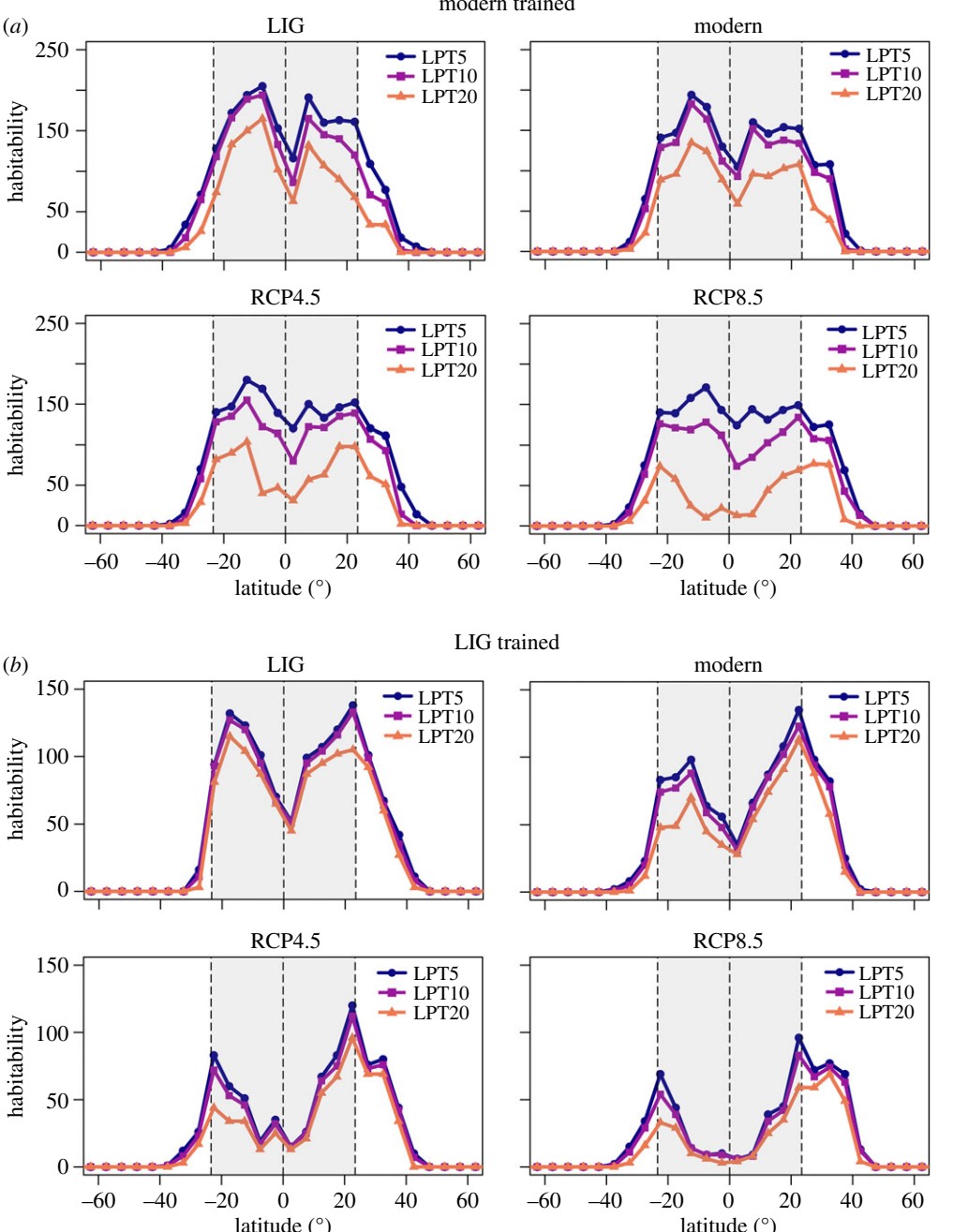

**Figure 2.** Latitudinal plots for predicted habitability from ecological niche modelling, using three binary thresholds: the 5th-, 10th- and 20th-percentiles of the lowest tail of habitat suitability values (5LPT, 10LPT and 20LPT). Plots indicate habitability for the LIG, modern, representative concentration pathway 4.5 (RCP4.5) and 8.5 (RCP8.5). In this instance, habitability is the sum of cells designated as habitable, within 5° latitudinal bins, by the mean prediction, trained on the (a) modern occurrence dataset, and (b) LIG occurrence dataset. Grey shaded area represents the extent of the tropics in the present day. Dashed lines indicate the Equator, the Northern Tropic and the Southern Tropic.

### 3.2.3. Range dynamics

The LIG binary maps (hindcasted and trained) indicate habitat suitability at equatorial latitudes (figures 2 and 3) for all binary thresholds. For latitudinal bin 0–5°, the LIG-trained ENM predicts 52, 51 and 45 presence cells for 5LPT, 10LPT and 20LPT, respectively, whereas the modern-trained ENM predicts 116, 86 and 63 cells (electronic supplementary material, SM3). Based on the results from the combined, modern-trained and LIG-trained projections, for all binary thresholds, habitability was generally greater within the tropics (−25° to 25°) during the LIG than in the modern (figure 2;

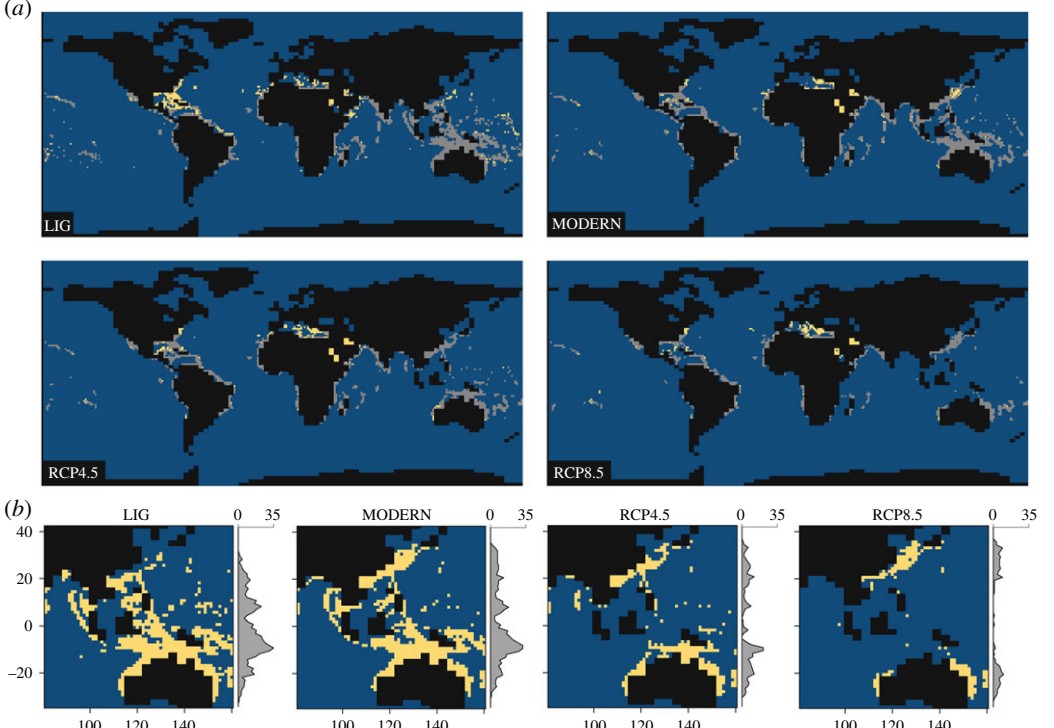

**Figure 3.** Combined binary maps (using threshold 20LPT) showing the potential distribution of reef corals under LIG, modern, RCP4.5 and RCP8.5 climate regimes. (*a*) Global maps of habitability under the four climate scenarios. Grey cells represent the potential distribution predicted through using only the modern reef coral training dataset. Yellow cells characterize cells gained when coupling LIG and modern-trained predictions. Blue cells indicate unsuitable locations, while black indicates no data cells (landmass). (*b*) Habitability maps clipped to Indo-Pacific for all climate scenarios to highlight change in habitability within the Coral Triangle. Yellow cells highlight suitable habitats, while blue cells indicate unsuitable locations, and black indicates no data cells (landmass).

electronic supplementary material, figures SM23 and SM25 and SM3). More specifically, results from the combined projections indicate that LIG equatorial regions (−5° to 5°) were more habitable than during the modern, with 34 (5LPT), 19 (10LPT) and 40 (20LPT) additional habitable cells predicted. In addition, habitability is generally greater at higher latitudes in both hemispheres for the LIG than in the modern, for all binary thresholds, suggesting that reef coral range size was likely larger during this time. To provide an example, for 5LPT, only three cells were predicted for latitudinal bin 40−45° in the modern Northern Hemisphere, yet 14 cells were predicted for the LIG (figure 2; electronic supplementary material, SM3).

Under RCP4.5 and RCP8.5, the models predict a considerable decrease in habitability (for all binary thresholds) within the tropics with a 5% (5LPT), 20% (10LPT) and 59% (20LPT) loss in habitable cells for RCP8.5 relative to the modern (SM3), notably in the centre of the hyper-diverse Coral Triangle (Indo-Pacific Ocean) (figure 3). By contrast, habitability increases significantly at higher latitudes (25−45°) in the Northern Hemisphere (figure 3), with a 36% (5LPT), 40% (10LPT) and 43% (20LPT) increase in habitable cells from the modern (SM3). South of the Equator, there is a similar gain at higher latitudes (25° to 45°), with a 30% (5LPT), 40% (10LPT) and 46% (20LPT) increase in habitable cells (SM3). However, the clamping masks, particularly for RCP8.5, highlight some uncertainty for equatorial tropical latitudes due to one environmental variable being outside the range used for model calibration (electronic supplementary material, SM2: figures SM21 and SM22). This finding highlights the need to incorporate fossil data from various climatic regimes to reduce uncertainty when projecting to future climate scenarios, which represent climate conditions outside modern experience [59].

## 4. Discussion

The LIG has occasionally been considered as an analogue for the impacts of global warming on species' distribution [13,60,61]. At face value, the LIG fossil record of reef corals depicts a global biogeographic

pattern that is considerably dissimilar to that observed in the present day (figure 1). Whereas today, both species abundance and diversity are highest within the tropics, with an equatorial dip, the LIG record shows a poleward shift in occurrence records and species diversity, with both increasing at higher latitudes in the Northern Hemisphere, and a prominent equatorial decline [24,25]. Support for this observed pattern is also provided when subsampling methods are applied to ameliorate biases pertaining to uneven latitudinal sampling [13]. However, using ENM, our findings provide little support for an equatorial decline in habitability for reef corals during the LIG. Instead, LIG reef corals occupied similar, though not equivalent, niches to those of modern reef corals. Our results suggest that during the LIG, global habitability was generally greater than the modern, particularly at equatorial latitudes. In addition, modelled habitability is more pronounced at high latitudes in both hemispheres, particularly, within the Northern Hemisphere. These results support a poleward range expansion of reef corals during the LIG, as suggested by previous studies [13,24], but not an equatorial decline [13]. This suggests that the observed equatorial decline in the fossil record might be a result of sampling bias within equatorial regions.

Under climate scenarios RCP4.5 and RCP8.5, our results indicate that high latitude habitability should increase (i.e. range expansion), while equatorial latitudes will see a large decrease in favourable habitats (i.e. range retraction) compared to the modern. Most notably, this includes the centre of the Coral Triangle, in the Indo-Pacific Ocean, the 'heart' of modern-day coral biodiversity, with 605 reef coral species, representing over three-quarters of global reef coral diversity [62]. In addition to the direct risk to biodiversity (including the multitude of species that live in reef coral environments), the economic, social and cultural benefits that coral reefs provide are of crucial importance to island communities within this region [63,64]. However, our results do also suggest that under RCP4.5 and RCP8.5, we can expect to see an increase in marginal habitats within the Coral Triangle, which might have important implications for future recovery. Nevertheless, it should also be borne in mind that reef corals, and coral reef ecosystems, are far less likely to survive other stressors, such as overfishing and pollution, when residing in marginal habitats. Furthermore, as our projections do not include aragonite saturation state, they may be relatively optimistic in the face of anthropogenic climate change as they do not account for the impacts of ocean acidification.

## 4.1. Implications for the evolution of the latitudinal biodiversity gradient

Attempts to reconstruct latitudinal biodiversity gradients in geological time have historically been achieved using direct reading of occurrence records [65], potentially leading to circumscribed results [66]. Although subsampling approaches have been applied in more recent studies, and provide more useful insights than uncorrected 'raw' diversity data [13], they ultimately cannot predict which data we might be missing, i.e. fossils and/or palaeoenvironments that we have not sampled, or that never entered the fossil record [67,68]. Our model results suggest that the LIG reef coral record is geographically biased through the amount of predicted habitable area within latitudinal bins, herein referred to as the latitudinal habitability gradient. Thus, we predict that, given suitable preservational conditions, we should expect to find additional LIG reef coral occurrence data from equatorial latitudes.

'Modern-type' latitudinal biodiversity gradients, with tropical peaks in diversity, have been suggested to form under cold icehouse climate regimes, whereas flattened gradients or temperate peaks are restricted to warmer greenhouse regimes [66]. Additionally, it has been proposed that non-modern-type latitudinal biodiversity gradients might also form during interglacials [66,69]. However, based on modelled habitability, our findings imply a contrarian result for the LIG, and suggest that interglacials might not be warm enough to produce latitudinal biodiversity gradients similar to those we observe in greenhouse climate regimes.

## 4.2. Utility of the fossil record in forecasting

Variable contribution analyses computed within biomod2 [55] clearly demonstrate that bathymetry and SST are the principal parameters controlling the distribution of both modern and LIG reef corals (electronic supplementary material, SM2: table SM3). It is to be expected that bathymetry has the largest explanative power as the very nature of reef (zooxanthellate) corals restricts them to shallow marine settings. While LIG bathymetry is generally similar to that of the modern at a global scale, several areas have experienced localized tectonic uplift (e.g. Huon Peninsula, Barbados, Sumba and Vanuatu [70]). However, this is not likely to impact our results, based on our sea-level sensitivity test (see electronic supplementary material, SM2). As both models used the same DEM, SST might be

considered the dominant factor in determining any difference between ENM and projected distributions. In fact, our results are in line with those suggested by palaeoclimatic proxy and modelling data, which propose that annual mean SST and seasonality at low latitudes for the LIG were similar to those of the modern [27,28,70]. Data from uplifted LIG coral reefs from the Huon Peninsula of New Guinea and western Australia also indicate mean annual temperatures and seasonal ranges comparable to present-day SST [70,71]. However, it has also been suggested that seasonality was greater at higher latitudes, with increased SST and insolation modelled in the Northern Hemisphere [27,72,73]. This might have permitted reef corals to reside at greater depths at higher latitudes, as well as at more poleward localities due to higher insolation and SST in the LIG, as supported by our habitability modelling results.

To avoid environmentally driven extinction, taxa need to be able to adapt to changing environments, either through distributional shifts (i.e. habitat tracking) or by altering their abiotic preferences [24,30,74]. The thermal limits of organisms are widely acknowledged as shaping diversity gradients on geological time scales (e.g. [69,75,76]). Although the latitudinal habitability gradient remained fairly stable between the LIG and modern climatic regimes, we might expect to see a greater reshaping of the latitudinal habitability and biodiversity gradients under future scenarios. While reef corals might be able to adapt over geological time intervals, it is unlikely that they will be able to keep pace with the rapid rate of climatic change predicted under RCP4.5 or RCP8.5 scenarios. Studies based solely on modern occurrences [16,77] propose a much more drastic decline in coral reefs under RCP4.5 and RCP8.5 scenarios than our results indicate for reef corals. This is likely the consequence of differences in methodological approach, selection of environmental variables, GCMs and differences in training datasets for modelling. Perhaps of particular note is the lack of aragonite saturation state and cumulative thermal stress variables applied in this study, whereas bathymetry was not directly considered in others (see [16]). We acknowledge that the inclusion of environmental variables such as aragonite saturation state would likely improve our model predictions; however, such data are not available for the geological record. With this in mind, our model outputs are likely optimistic in the face of anthropogenic climate change, particularly for reef corals in high latitude regions, as the aragonite saturation state is generally lower than at tropical latitudes [78]. In relation to other environmental variables, best practice requires the construction of generalized models when projecting to other time slices. An ENM focused on one spatial or temporal extent may perform very well with a large number of explanatory variables. However, an ENM might also find causal relationships that are non-existent, resulting in over-fitting, hindering the applicability and any projections of the model under different spatial/temporal extents. Sensitivity analyses comparing modern projections from modern-trained ENMs with and without aragonite saturation state suggest little difference between predictions at global scale. However, some noticeable variations are observable at local scale (electronic supplementary material, SM2: figure SM2). Our variable contribution analyses indicate aragonite saturation state is of secondary importance in the construction of global ENMs (electronic supplementary material, SM2: table SM1). However, under continued acidification of the world's oceans, aragonite saturation state may become more of a limiting factor for reef corals than what is observed in the modern. Therefore, RCP4.5 and RCP8.5 projections should be considered as relatively optimistic in the face of continued $CO_2$ emissions. However, this is conditional on reef corals being able to acclimatize to heightened temperatures or shift distributions to suitable temperature localities in the first place. Integrating palaeontological data into ENM promotes two clear advantages to aid model projections: (i) the opportunity to test models with empirical evidence across temporal windows and climate regimes and (ii) the ability to combine palaeontological and neontological data to consider the full ecological limits of a particular taxon, rather than just the occupied niche in its current biogeographic pattern.

Although the LIG fossil record is biased, the inclusion of fossil reef coral data provides additional insight into the niche hypervolume and potential distribution of reef corals under future climate change. As with previous studies [20], predictions including fossil data suggest significantly more habitability at wider geographical ranges than those based solely on modern occurrences. ENM predictions which only include modern occurrences run the risk of modelling the occupied niche, whereas fossil occurrences can provide additional understanding into the fundamental niche of a species [79,80]. Indeed, this particular study still might provide limited insights into the fundamental niche due to the incompleteness of the LIG reef coral record. However, the inclusion of fossil occurrences enables us to test which models perform best across temporal boundaries, as well as the importance of establishing the fundamental niche using true positive fraction. It is clear from our results that a multi-temporal approach improves predictive performance, particularly when

considering higher binary thresholds, providing support for projected habitat suitability under future climate scenarios.

# 5. Conclusion

Our results suggest that LIG reef corals might have expanded their poleward ranges during the LIG, but that they did not experience an equatorial decline. Instead, the apparent equatorial decline appears to reflect an aspect of bias (i.e. absence of reported fossil collections/occurrences) that cannot currently be ameliorated by existing subsampling approaches. Our findings are in agreement with the results from multiple modelling and palaeoclimatic studies, which suggest that annual mean SST and seasonal ranges at equatorial latitudes in the LIG were comparable to today [27,28,70]. Under the future (2100 AD) climate scenarios RCP4.5 and RCP8.5, our results suggest a relatively large decrease in habitat suitability at equatorial latitudes, and an increase at higher latitudes, particularly within the Northern Hemisphere. However, our conclusions are likely somewhat optimistic in the face of anthropogenic climate change as they do not account for ocean acidification, destructive fishing practices or other anthropogenic environmental impacts, and yet they make alarming predictions. Finally, the fossil record provides additional insights into the niche hypervolume of reef corals, and improves predictive performance. Consequently, the fossil record might improve biodiversity risk assessments under future climate change and it should therefore play an important role in focusing conservation efforts.

Data accessibility. Data are available from the Dryad Digital Repository: https://doi.org/10.5061/dryad.2c0g95d [81].

Authors' contributions. L.A.J. and P.D.M. conceived and designed the project; L.A.J. performed the analyses and interpretation of data; P.J.V. provided the Last Interglacial climate data; S.-J.K. (Getech Group Plc) provided the global digital elevation model; L.A.J., P.D.M., A.F. and P.A.A. contributed to the writing of the manuscript; L.A.J. produced the figures.

Competing interests. We declare we have no competing interests.

Funding. L.A.J. was supported by an Imperial College London President's PhD Scholarship. P.D.M.'s contribution was supported by a Royal Society University Research Fellowship (UF160216).

Acknowledgements. We are grateful for the efforts of all those who have collected modern-day and Pleistocene reef coral data, as well as those who have entered these data into the Ocean Biogeographic Information System and the Paleobiology Database (in particular Wolfgang Kiessling). We would also like to thank Getech Group Plc for supplying the digital elevation model used in this study. Helpful discussions with Neftalí Sillero (University of Porto), Elena Couce (Centre for Environment, Fisheries and Aquaculture Science) and Alessandro Chiarenza (Imperial College London) improved this work, as did comments from six anonymous reviewers. This is Paleobiology Database official publication number 340.

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
