## [Reviewer comments · Royal Society Open Science]

Review History

RSOS-182111.R0 (Original submission)

Review form: Reviewer 1

Is the manuscript scientifically sound in its present form?

Yes

Are the interpretations and conclusions justified by the results?

Yes

Is the language acceptable?

Yes

Is it clear how to access all supporting data?

Yes

Do you have any ethical concerns with this paper?

No

Have you any concerns about statistical analyses in this paper?

I do not feel qualified to assess the statistics

Recommendation?

Accept with minor revision (please list in comments)

Comments to the Author(s)

This paper is a nice analysis of zooxanthellate coral (Z-coral) biogeography using Ecological Niche Modelling (ENM). We like the author's use of the ENM to assess the completeness of the fossil record as well as their future predictions. It is clear that this manuscript has already been through several rounds of revision and so should be accepted once a few additional things are addressed.

Moderate/Major Issues

1. "Modern" baselines. What do you define as "modern"? Is it present data (i.e., 2018/2019), data from the preindustrial era, or some other time? This might seem like splitting hairs, but the climate (and thus environment) is changing so fast that coral reef distribution may not be representative of current settings, but rather the averages from 20 years ago, or 100 years ago. It should be explicitly stated in the methods what you mean by "modern" and better yet, would be to see if your ENM data change significantly (or are at least different) if you use 2018 data vs. 2000 data vs. pre-industrial data. These results would be interesting in their own rights and would be a good test for a) the sensitivity of the ENM to input data and b) the shifting baselines that reefs are currently subjected to. If this has already been tested sufficiently by other groups, it would also be ok to just cite those works.

2. Saturation State. Other reviewers have already pointed this issue out, but we still do not feel it has been properly addressed. The authors themselves state (correctly) that SST is the primary control on Z-coral reefs, but that light availability, saturation state (Ω), salinity and nutrient levels are also important (lines 144-147). Yes, they are "secondary factors" and thus not as important as SST, but as the ENM shows, several of these factors are still important environmental variables (lines 138-140). While it is understood that saturation state is not recorded directly in the fossil record, most of the variables are generated by a climate model anyway. At the very least it would be good to incorporate modern saturation states to see if they influence the ENM. Feely et al., 2009 (<https://doi.org/10.5670/oceanog.2009.95>) have done a nice job of reconstructing modern, pre-industrial, and future saturation states that could be used (and there are probably better models that have come out in the last decade). If this test shows that Ω values do not change the ENM, then they can ignore it for the past... but if they change, then the authors should at least highlight that this is an unknown parameter that should be included in future models, especially since many GCMs are not including carbonate chemistry.

3. Ecological variables. Many of the ecological variables discussed are quite variable within a $1.25^\circ \times 1.25^\circ$ grid cell (e.g., depth or salinity can change dramatically on a shelf edge reef vs. lagoonal reef). These factors can have significant effects on a reef ecosystem. At the very least, the authors should note that this local variability will not be captured by their ENM and would be significant. Better yet would be to do the ENM at a higher resolution for some significant regions (e.g., the coral triangle), but the model or computing power may be limited. Also, what about seasonality or temperature extremes? Some papers have shown that temperature extremes might be linked with adaptability (e.g., Palumbi et al., 2014; DOI: 10.1126/science.1251336).

Minor issues:

- There were too many acronyms (especially similar acronyms) and this often made the text difficult to understand. Please remove unnecessary acronyms (e.g., LBG, LHG), leaving only the common, or highly used ones (e.g., SST, DEM, LIG). A good general rule is to limit acronyms to ~3 – 5 in a paper/grant).
- Throughout the text there are places where comparative adjectives are used (e.g., “larger” in line 245, “more favorable” in line 261, etc.) but there is only one thing in the sentence. Please go through the text and make sure that if you are using a comparative adjective to compare 2 things, it is clear what both of those things are (i.e., larger than what?).
- Line 15-17: The last interglacial is not the fossil record. We suggest rephrasing to: “The paleontological record from the Last Interglacial (LIG; 125,000 years ago), a time of global warming, suggests that reef corals experienced poleward range shifts and an equatorial decline relative to their modern distribution.”
- Line 70: missing punctuation in the brackets
- Line 84: Please define what is meant by “ecotype-level”. It is good to be explicit in the methods section.
- Lines 95-97: Suggest you include an explicit statement saying that the zooxanthellate characteristics are assumed to be the same as the modern for the fossil coral taxa (obvious yes, but an assumption nonetheless). For example: Kiessling and Kocsis 2015, doi: 10.1017/pab.2015.6 ; Swain et al 2018, doi: 10.3354/meps12445)
- Line 133: saturation state/alkalinity, etc. are also limiting factors.
- Line 186: semicolon not needed
- Line 305: What might explain the equatorial decline? If not the factors analyzed in the ENM, might there be something else?
- Lines 315-316: It would also be good to mention that stressed coral reef ecosystems are a lot less likely to survive other stressors, such as pollution, overfishing etc.
- Line 338: Please define/explain Variable Contribution analysis (or provide a citation)
- Lines 338-341: Bathymetric variability should also be discussed here, either in terms of the lack of ENM resolution and the tectonic changes that have occurred; i.e., LIG bathymetry is generally similar to modern bathymetry, but many places (e.g., Jamaca or Vanuatu) have experienced non-trivial uplift due to tectonics.
- Line 389: replace “(i.e. absence)” with “(i.e. absence of reported fossil collections/occurrences)”
- Line 406: Suggest adding “anthropogenic” before “environmental impacts”

Figure Edits

*Please make the below corrections in the supplemental material as well.

- Fig. 1: Please define “Occ.” In the figure caption.

- Figs. 1 & 2: Define the dashed lines and greyed area in the plots (presumably it is the tropics/equator but the reader should not need to guess).
- Fig. 2: Please define what the binary thresholds in the caption mean as well as the different scenarios (i.e., LIG, RCP4.5). Figures should be able to stand alone.
- Fig. 3: These plots are definitely difficult to read, the blue should be lightened... it is almost impossible to distinguish blue cells from black cells unless you're in perfect lighting conditions. It's great that you have checked for colorblindness, but it just looks like some yellow and grey dots in black squares to someone of normal vision. Since this is the second time a reviewer has highlighted issue, you should change the colors.

Review form: Reviewer 2 (Carl Simpson)

Is the manuscript scientifically sound in its present form?

Yes

Are the interpretations and conclusions justified by the results?

Yes

Is the language acceptable?

Yes

Is it clear how to access all supporting data?

Yes

Do you have any ethical concerns with this paper?

No

Have you any concerns about statistical analyses in this paper?

No

Recommendation?

Accept with minor revision (please list in comments)

Comments to the Author(s)

In this paper, the authors use modern and paleontological (Last interglacial [LIG]) occurrence data to niche model the distributions of reef corals. Using these niche models, they then predict the response to two future climate scenarios. The goal of this paper is to predict how corals will respond to future climate change. A model of coral distributions is needed in order to make this prediction. And the authors use fossil and modern distributions to train their model.

The paper has three stated goals:

“Here, we use ENM to: (1) evaluate the extent to which the LIG equatorial decline of reef corals is the result of fossil bias or a genuine loss in habitability; (2) quantify the variation in geographic range of reef corals during the LIG and under future climate scenarios, with implications for understanding the evolution of the reef coral latitudinal biodiversity gradient; and (3)

demonstrate the value of utilising fossil reef coral data to improve biodiversity risk assessment through consideration of the realised niche under varied climate states.”

All of these goals address important questions. As far as I know, goal 2 and 3 are novel and this paper provides important results. Goal 1 was previously explored by Kiessling et al (2012) [which I was a coauthor].

All in all, I think that, although this paper tackles an important issue, it does not meet its potential because it is confusing. The focus on occurrences and ENM methods and not the questions posed is a big part of this confusion for me. Take for example, the LIG and modern diversity gradients. Although these are major parts of the the paper (goals 1 and 2), no where is diversity shown, only numbers of occurrences and numbers of habitable cells.

Major comments:

I understand the motive for studying the relationships between coral distributions and climate. I don't think a convincing case for using niche models is made in the introduction. No doubt there are sampling issues with the past that cannot be fully ameliorated by sub-sampling. But framing the current paper as a methodological correction to Kiessling et al limits the current paper's appeal and impact. ENM are needed for a much more important reason than correcting Kiessling – a niche model is necessary for the forward prediction into future climate scenarios.

I suggest adding in a paragraph into the introduction, at about line 73, to point out that there is no other way to make a prediction into the future than with a well trained model. Adding in fossil occurrences from the last interglacial, when the earth was warmer than today, aids in the modeling effort because modern occurrences alone sample a small range of possible climatic variation. More sampled variation in temperatures and occurrences will lead to a higher fidelity model.

Geographic ranges surely are species specific, therefore I can't see how lumping occurrences of all coral species together can say anything about the range shifts from LIG to the modern. However, those ENMs are the only option for predicting the coral occurrences in the future. Would species specific ENMs be too noisy?

I find understanding Figure 2 to be confusing. There are many combinations of training dataset and modeled pattern that are hard to tease apart. For one, I'm surprised that the modern-trained modern habitability is so different from the observed modern occurrences. The peak habitability in the southern hemisphere is 5 or more degrees off of the peak occurrences in the raw data. This mismatch makes me wonder if the number of habitable cells is informative for diversity. What is the pattern of latitudinal diversity in this dataset anyway?

Decision letter (RSOS-182111.R0)

07-Mar-2019

Dear Mr Jones

On behalf of the Editors, I am pleased to inform you that your Manuscript RSOS-182111 entitled "Coupling of palaeontological and neontological reef coral data improves forecasts of biodiversity responses under global climatic change" has been accepted for publication in Royal Society Open

Science subject to minor revision in accordance with the referee suggestions. Please find the referees' comments at the end of this email.

The reviewers and handling editors have recommended publication, but also suggest some minor revisions to your manuscript. Therefore, I invite you to respond to the comments and revise your manuscript.

- Ethics statement

- Data accessibility

If you wish to submit your supporting data or code to Dryad (<http://datadryad.org/>), or modify your current submission to dryad, please use the following link:
<http://datadryad.org/submit?journalID=RSOS&manu=RSOS-182111>

- Competing interests

- Authors' contributions

- Acknowledgements

- Funding statement

Because the schedule for publication is very tight, it is a condition of publication that you submit the revised version of your manuscript before 16-Mar-2019. Please note that the revision deadline will expire at 00.00am on this date. If you do not think you will be able to meet this date please let me know immediately.

Supplementary files will be published alongside the paper on the journal website and posted on the online figshare repository (<https://rs.figshare.com/>). The heading and legend provided for each supplementary file during the submission process will be used to create the figshare page,

so please ensure these are accurate and informative so that your files can be found in searches. Files on figshare will be made available approximately one week before the accompanying article so that the supplementary material can be attributed a unique DOI.

on behalf of Professor Stephen Hesselbo (Associate Editor) and Professor Jon Blundy (Subject Editor)
openscience@royalsociety.org

Associate Editor Comments to Author (Professor Stephen Hesselbo):

Associate Editor: 1

Comments to the Author:

The manuscript has now been thoroughly reviewed and the reviewers have made a number of important points that, if addressed, will significantly improve the impact of the eventual published version. I agree with all the comments and suggestions made by the reviewers and encourage the authors to consider them carefully.

Reviewer comments to Author:

Reviewer: 1

Comments to the Author(s)

This paper is a nice analysis of zooxanthellate coral (Z-coral) biogeography using Ecological Niche Modelling (ENM). We like the author's use of the ENM to assess the completeness of the fossil record as well as their future predictions. It is clear that this manuscript has already been through several rounds of revision and so should be accepted once a few additional things are addressed.

Moderate/Major Issues

1. "Modern" baselines. What do you define as "modern"? Is it present data (i.e., 2018/2019), data from the preindustrial era, or some other time? This might seem like splitting hairs, but the

climate (and thus environment) is changing so fast that coral reef distribution may not be representative of current settings, but rather the averages from 20 years ago, or 100 years ago. It should be explicitly stated in the methods what you mean by “modern” and better yet, would be to see if your ENM data change significantly (or are at least different) if you use 2018 data vs. 2000 data vs. pre-industrial data. These results would be interesting in their own rights and would be a good test for a) the sensitivity of the ENM to input data and b) the shifting baselines that reefs are currently subjected to. If this has already been tested sufficiently by other groups, it would also be ok to just cite those works.

2. Saturation State. Other reviewers have already pointed this issue out, but we still do not feel it has been properly addressed. The authors themselves state (correctly) that SST is the primary control on Z-coral reefs, but that light availability, saturation state (Ω), salinity and nutrient levels are also important (lines 144-147). Yes, they are “secondary factors” and thus not AS important as SST, but as the ENM shows, several of these factors are still important environmental variables (lines 138-140). While it is understood that saturation state is not recorded directly in the fossil record, most of the variables are generated by a climate model anyway. At the very least it would be good to incorporate modern saturation states to see if they influence the ENM. Feely et al., 2009 (<https://doi.org/10.5670/oceanog.2009.95>) have done a nice job of reconstructing modern, pre-industrial, and future saturation states that could be used (and there are probably better models that have come out in the last decade). If this test shows that Ω values do not change the ENM, then they can ignore it for the past... but if they change, then the authors should at least highlight that this is an unknown parameter that should be included in future models, especially since many GCMs are not including carbonate chemistry.

3. Ecological variables. Many of the ecological variables discussed are quite variable within a 1.25° x 1.25° grid cell (e.g., depth or salinity can change dramatically on a shelf edge reef vs. lagoonal reef). These factors can have significant effects on a reef ecosystem. At the very least, the authors should note that this local variability will not be captured by their ENM and would be significant. Better yet would be to do the ENM at a higher resolution for some significant regions (e.g., the coral triangle), but the model or computing power may be limited. Also, what about seasonality or temperature extremes? Some papers have shown that temperature extremes might be linked with adaptability (e.g., Palumbi et al., 2014; DOI: 10.1126/science.1251336).

Minor issues:

- There were too many acronyms (especially similar acronyms) and this often made the text difficult to understand. Please remove unnecessary acronyms (e.g., LBG, LHG), leaving only the common, or highly used ones (e.g., SST, DEM, LIG). A good general rule is to limit acronyms to ~3 – 5 in a paper/grant).
- Throughout the text there are places where comparative adjectives are used (e.g., “larger” in line 245, “more favorable” in line 261, etc.) but there is only one thing in the sentence. Please go through the text and make sure that if you are using a comparative adjective to compare 2 things, it is clear what both of those things are (i.e., larger than what?).
- Line 15-17: The last interglacial is not the fossil record. We suggest rephrasing to: “The paleontological record from the Last Interglacial (LIG; 125,000 years ago), a time of global warming, suggests that reef corals experienced poleward range shifts and an equatorial decline relative to their modern distribution.”
- Line 70: missing punctuation in the brackets

- Line 84: Please define what is meant by “ecotype-level”. It is good to be explicit in the methods section.
- Lines 95-97: Suggest you include an explicit statement saying that the zooxanthellate characteristics are assumed to be the same as the modern for the fossil coral taxa (obvious yes, but an assumption nonetheless). For example: Kiessling and Kocsis 2015, doi: 10.1017/pab.2015.6 ; Swain et al 2018, doi: 10.3354/meps12445)
- Line 133: saturation state/alkalinity, etc. are also limiting factors.
- Line 186: semicolon not needed
- Line 305: What might explain the equatorial decline? If not the factors analyzed in the ENM, might there be something else?
- Lines 315-316: It would also be good to mention that stressed coral reef ecosystems are a lot less likely to survive other stressors, such as pollution, overfishing etc.
- Line 338: Please define/explain Variable Contribution analysis (or provide a citation)
- Lines 338-341: Bathymetric variability should also be discussed here, either in terms of the lack of ENM resolution and the tectonic changes that have occurred; i.e., LIG bathymetry is generally similar to modern bathymetry, but many places (e.g., Jamaca or Vanuatu) have experienced non-trivial uplift due to tectonics.
- Line 389: replace “(i.e. absence)” with “(i.e. absence of reported fossil collections/occurrences)”
- Line 406: Suggest adding “anthropogenic” before “environmental impacts”

Figure Edits

*Please make the below corrections in the supplemental material as well.

- Fig. 1: Please define “Occ.” In the figure caption.
- Figs. 1 & 2: Define the dashed lines and greyed area in the plots (presumably it is the tropics/equator but the reader should not need to guess).
- Fig. 2: Please define what the binary thresholds in the caption mean as well as the different scenarios (i.e., LIG, RCP4.5). Figures should be able to stand alone.
- Fig. 3: These plots are definitely difficult to read, the blue should be lightened... it is almost impossible to distinguish blue cells from black cells unless you’re in perfect lighting conditions. It’s great that you have checked for colorblindness, but it just looks like some yellow and grey dots in black squares to someone of normal vision. Since this is the second time a reviewer has highlighted issue, you should change the colors.

Reviewer: 2

Comments to the Author(s)

In this paper, the authors use modern and paleontological (Last interglacial [LIG]) occurrence data to niche model the distributions of reef corals. Using these niche models, they then predict the response to two future climate scenarios. The goal of this paper is to predict how corals will

respond to future climate change. A model of coral distributions is needed in order to make this prediction. And the authors use fossil and modern distributions to train their model.

The paper has three stated goals:

“Here, we use ENM to: (1) evaluate the extent to which the LIG equatorial decline of reef corals is the result of fossil bias or a genuine loss in habitability; (2) quantify the variation in geographic range of reef corals during the LIG and under future climate scenarios, with implications for understanding the evolution of the reef coral latitudinal biodiversity gradient; and (3) demonstrate the value of utilising fossil reef coral data to improve biodiversity risk assessment through consideration of the realised niche under varied climate states.”

All of these goals address important questions. As far as I know, goal 2 and 3 are novel and this paper provides important results. Goal 1 was previously explored by Kiessling et al (2012) [which I was a coauthor].

All in all, I think that, although this paper tackles an important issue, it does not meet its potential because it is confusing. The focus on occurrences and ENM methods and not the questions posed is a big part of this confusion for me. Take for example, the LIG and modern diversity gradients. Although these are major parts of the the paper (goals 1 and 2), no where is diversity shown, only numbers of occurrences and numbers of habitable cells.

Major comments:

I understand the motive for studying the relationships between coral distributions and climate. I don't think a convincing case for using niche models is made in the introduction. No doubt there are sampling issues with the past that cannot be fully ameliorated by sub-sampling. But framing the current paper as a methodological correction to Kiessling et al limits the current paper's appeal and impact. ENM are needed for a much more important reason than correcting Kiessling – a niche model is necessary for the forward prediction into future climate scenarios.

I suggest adding in a paragraph into the introduction, at about line 73, to point out that there is no other way to make a prediction into the future than with a well trained model. Adding in fossil occurrences from the last interglacial, when the earth was warmer than today, aids in the modeling effort because modern occurrences alone sample a small range of possible climatic variation. More sampled variation in temperatures and occurrences will lead to a higher fidelity model.

Geographic ranges surely are species specific, therefore I can't see how lumping occurrences of all coral species together can say anything about the range shifts from LIG to the modern. However, those ENMs are the only option for predicting the coral occurrences in the future. Would species specific ENMs be too noisy?

I find understanding Figure 2 to be confusing. There are many combinations of training dataset and modeled pattern that are hard to tease apart. For one, I'm surprised that the modern-trained modern habitability is so different from the observed modern occurrences. The peak habitability in the southern hemisphere is 5 or more degrees off of the peak occurrences in the raw data. This mismatch makes me wonder if the number of habitable cells is informative for diversity. What is the pattern of latitudinal diversity in this dataset anyway?

Author's Response to Decision Letter for (RSOS-182111.R0)

See Appendices A & B.

Decision letter (RSOS-182111.R1)

01-Apr-2019

Dear Mr Jones,

I am pleased to inform you that your manuscript entitled "Coupling of palaeontological and neontological reef coral data improves forecasts of biodiversity responses under global climatic change" is now accepted for publication in Royal Society Open Science.

on behalf of Professor Stephen Hesselbo (Associate Editor) and Professor Jon Blundy (Subject Editor)
openscience@royalsociety.org

Associate Editor Comments to Author (Professor Stephen Hesselbo):
The authors have carried out a thorough review and have addressed all the reviewers' comments.

Appendix A

Imperial College
London

Lewis A. Jones
Dept. Earth Science & Engineering
Imperial College London
London, SW7 2AZ, UK

Email: l.jones16@imperial.ac.uk

12 April 2019

Dear Editor,

Below, we respond to each individual point raised by the associate editor and reviewers. Our modifications to the original manuscript are attached in a tracked format, and a “clean” version is also submitted. Comments from the referees herein are in *italics*, while our responses are in **bold**.

The main results and conclusions from our original submission are unchanged following this revision. We hope that these changes and responses will satisfy the associate editor and reviewers, and that our MS can now be accepted for publication.

Yours sincerely,

Lewis A. Jones

Associate Editor Comments to Author (Professor Stephen Hesselbo):

Associate Editor: 1

Comments to the Author:

The manuscript has now been thoroughly reviewed and the reviewers have made a number of important points that, if addressed, will significantly improve the impact of the eventual published version. I agree with all the comments and suggestions made by the reviewers and encourage the authors to consider them carefully.

First of all, we would like to thank the associate editor for their involvement with this manuscript. We hope that our responses below satisfy their concerns, as well as those of the reviewers.

Reviewer comments to Author:

Reviewer: 1

To begin with, we would like to thank reviewer 1 for their time, constructive criticism and clear efforts to improve this manuscript. We have taken their comments wholeheartedly on board and have made several amendments to our work based on their insight. Please see details below.

Comments to the Author(s)

This paper is a nice analysis of zooxanthellate coral (Z-coral) biogeography using Ecological Niche Modelling (ENM). We like the author's use of the ENM to assess the completeness of the fossil record as well as their future predictions. It is clear that this manuscript has already been through several rounds of revision and so should be accepted once a few additional things are addressed.

Moderate/Major Issues

1. *“Modern” baselines. What do you define as “modern”? Is it present data (i.e., 2018/2019), data from the preindustrial era, or some other time? This might seem like splitting hairs, but the climate (and thus environment) is changing so fast that coral reef distribution may not be representative of current settings, but rather the averages from 20 years ago, or 100 years ago. It should be explicitly stated in the methods what you mean by “modern” and better yet, would be to see if your ENM data change significantly (or are at least different) if you use 2018 data vs. 2000 data vs. pre-industrial data. These results would be interesting in their own rights and would be a good test for a) the sensitivity of the ENM to input data and b) the shifting baselines that reefs are currently subjected to. If this has already been tested sufficiently by other groups, it would also be ok to just cite those works.*

We have adjusted our text accordingly to specify that the modern data is of pre-industrial climate conditions. Whilst we agree with the reviewers that it would be interesting to essentially use a time-calibrated ENM, this would be a study or perhaps even two in its own right. It would require filtering all occurrences to specific collection times and relating it to climatological means for that date. This would be a substantial undertaking and is not within the scope of this particular study. What we have done here is use pre-industrial climate conditions as it represents the long-term climate which resulted into the biogeographic pattern we observe today.

2. *Saturation State. Other reviewers have already pointed this issue out, but we still do not feel it has been properly addressed. The authors themselves state (correctly) that SST is the primary control on Z-coral reefs, but that light availability, saturation state (Ω), salinity and nutrient levels are also important (lines 144-147). Yes, they are “secondary factors” and thus not AS important as SST, but as the ENM shows, several of these factors are still important environmental variables (lines 138-140). While it is understood that saturation state is not recorded directly in the fossil record, most of the variables are generated by a climate model anyway. At the very least it would be good to incorporate modern saturation states to see if they influence the ENM. Feely et al., 2009 (<https://doi.org/10.5670/oceanog.2009.95>) have done a nice job of reconstructing modern, pre-industrial, and future saturation states that could be used (and there are probably better models that have come out in the last decade). If this test shows that Ω values do not change the ENM, then they can ignore it for the past... but if they change, then the authors should at least highlight that this is an unknown parameter that should be included in future models, especially since many GCMs are not including carbonate chemistry.*

To address this issue we have now generated a modern-trained ENM with aragonite saturation state (Ω) included. As a sensitivity test we generated difference maps of the geographic projections for the modern from ENMs with and without Ω included. We also ran variable contribution tests to see the importance of Ω in our ENMs. We found that at a global scale, predictions are essentially the same, however, there are some differences at a few isolated coastlines. In terms of variable contribution, our analyses confirms that Ω is indeed of secondary significance, with similar contributions to the ENM as irradiance variables. As the reviewer notes and our results confirm, although Ω may not be as important as bathymetry and sea surface temperature, it is not insignificant. Throughout the MS we have now tailored the text so it is in line with these results.

3. *Ecological variables. Many of the ecological variables discussed are quite variable within a 1.25° x 1.25° grid cell (e.g., depth or salinity can change dramatically on a shelf edge reef vs. lagoonal reef). These factors can have significant effects on a reef ecosystem. At the very least, the authors should note that this local variability will not be captured by their ENM and would be significant. Better yet would be to do the ENM at a higher resolution for some significant regions (e.g., the coral triangle), but the model or computing power may be limited. Also, what about seasonality or temperature extremes? Some papers have shown that temperature extremes might be linked with adaptability (e.g., Palumbi et al., 2014; DOI: 10.1126/science.1251336).*

We raise this issue in Supplementary Material 2 (first paragraph), particularly in regards to salinity. However, we have added an additional couple of lines expanding on this. Whilst we agree with the reviewer that it may have significant impacts at localised scales, the spatial scale needed to capture such impacts would be on the scale of metres, opposed to kilometres. Unfortunately, this level of resolution for environmental data at a global scale is simply not available at this time. There are several works using remotely operated vehicles to capture data at this resolution (particularly on cold-water corals), but as far as we are aware, these usually focus on localised transects. We hope to see higher-resolution data from general circulation models in the future which can accurately account for climate at higher resolution. However, it should also be considered that there may be a higher level of uncertainty associated with future or past climate predictions at such resolutions, and available computing power is restricted. In regards to the last point concerning acclimatization, it is correct that the historical climate experience of some reef coral populations makes them more adaptable to temperature extremes (even within the same species). However, our models treat reef corals as a single entity (ecotype), and whilst we could introduce seasonal variables into our models, it might have significant impacts on any model projections to other time-slices due to overfitting and multicollinearity. A mechanistic ENM approach would be much more beneficial for the type of analyses reviewer 1 is suggesting, rather than a correlative one.

Minor issues:

- *There were too many acronyms (especially similar acronyms) and this often made the text difficult to understand. Please remove unnecessary acronyms (e.g., LBG, LHG), leaving only the common, or highly used ones (e.g., SST, DEM, LIG). A good general rule is to limit acronyms to ~3 – 5 in a paper/grant).*

We have taken the reviewers concern on board and removed both the acronyms LBG and LHG.

- *Throughout the text there are places where comparative adjectives are used (e.g., “larger” in line 245, “more favorable” in line 261, etc.) but there is only one thing in the sentence. Please go through the text and make sure that if you are using a comparative adjective to compare 2 things, it is clear what both of those things are (i.e., larger than what?).*

Amended, thank you.

- *Line 15-17: The last interglacial is not the fossil record. We suggest rephrasing to: “The paleontological record from the Last Interglacial (LIG; 125,000 years ago), a time of global warming, suggests that reef corals experienced poleward range shifts and an equatorial decline relative to their modern distribution.”*

Amended, thank you for the suggestion.

- *Line 70: missing punctuation in the brackets*

Amended, thank you.

- *Line 84: Please define what is meant by “ecotype-level”. It is good to be explicit in the methods section.*

Added.

- *Lines 95-97: Suggest you include an explicit statement saying that the zooxanthellate characteristics are assumed to be the same as the modern for the fossil coral taxa (obvious yes, but an assumption nonetheless). For example: Kiessling and Kocsis 2015, doi: 10.1017/pab.2015.6 ; Swain et al 2018, doi: 10.3354/meps12445)*

We have added the following statement within this paragraph “Assignment of zooxanthellate class focused on the principle of uniformitarianism for fossil coral taxa.”

- *Line 133: saturation state/alkalinity, etc. are also limiting factors.*

Whilst we agree, we are specifically referring to those variables that can be determined for the fossil record from currently available GCM outputs (second half of the sentence).

- *Line 186: semicolon not needed*

Removed, thank you.

- *Line 305: What might explain the equatorial decline? If not the factors analyzed in the ENM, might there be something else?*

We have added: “This suggests that the observed equatorial decline in the fossil record might be a result of sampling bias within equatorial regions.”

- *Lines 315-316: It would also be good to mention that stressed coral reef ecosystems are a lot less likely to survive other stressors, such as pollution, overfishing etc.*

Thank you for raising this important point. We have added: “Nevertheless, it should also be borne in mind that reef corals, and coral reef ecosystems, are far less likely to survive other stressors, such as overfishing and pollution, when residing in marginal habitats. “

- *Line 338: Please define/explain Variable Contribution analysis (or provide a citation)*

Added computed within biomod2 and citation for reference.

- *Lines 338-341: Bathymetric variability should also be discussed here, either in terms of the lack of ENM resolution and the tectonic changes that have occurred; i.e., LIG bathymetry is generally similar to modern bathymetry, but many places (e.g., Jamaica or Vanuatu) have experienced non-trivial uplift due to tectonics.*

Added two sentences within the paragraph on this matter: “Whilst LIG bathymetry is generally similar to that of the modern at a global scale, several areas have experienced localised tectonic uplift (e.g. Huon Peninsula, Barbados, Sumba and Vanuatu (McCulloch and Esat, 2000)). However, this is not likely to impact our results, based on our sea-level sensitivity test (see supplementary material 2).”

- *Line 389: replace “(i.e. absence)” with “(i.e. absence of reported fossil collections/occurrences)”*

Amended.

- *Line 406: Suggest adding “anthropogenic” before “environmental impacts”*

Amended.

Figure Edits

**Please make the below corrections in the supplemental material as well.*

- *Fig. 1: Please define “Occ.” In the figure caption.*

Implemented.

- *Figs. 1 & 2: Define the dashed lines and greyed area in the plots (presumably it is the tropics/equator but the reader should not need to guess).*

Implemented.

- *Fig. 2: Please define what the binary thresholds in the caption mean as well as the different scenarios (i.e., LIG, RCP4.5). Figures should be able to stand alone.*

Implemented.

- *Fig. 3: These plots are definitely difficult to read, the blue should be lightened... it is almost impossible to distinguish blue cells from black cells unless you’re in perfect lighting conditions. It’s great that you have checked for colorblindness, but it just looks like some yellow and grey dots in black squares to someone of normal vision. Since this is the second time a reviewer has highlighted issue, you should change the colors.*

Implemented.

Reviewer: 2

Comments to the Author(s)

In this paper, the authors use modern and paleontological (Last interglacial [LIG]) occurrence data to niche model the distributions of reef corals. Using these niche models, they then predict the response to two future climate scenarios. The goal of this paper is to predict how corals will respond to future climate change. A model of coral distributions is needed in order to make this prediction. And the authors use fossil and modern distributions to train their model.

To begin with, we would like to thank reviewer 2 for their time, constructive criticism and clear efforts to improve this manuscript. We have taken their comments wholeheartedly on board and have made several amendments to our work based on their insight. Please see details below.

The paper has three stated goals:

“Here, we use ENM to: (1) evaluate the extent to which the LIG equatorial decline of reef corals is the result of fossil bias or a genuine loss in habitability; (2) quantify the variation in geographic range of reef corals during the LIG and under future climate scenarios, with implications for understanding the evolution of the reef coral latitudinal biodiversity gradient; and (3) demonstrate the value of utilising fossil reef coral data to improve biodiversity risk assessment through consideration of the realised niche under varied climate states.”

All of these goals address important questions. As far as I know, goal 2 and 3 are novel and this paper provides important results. Goal 1 was previously explored by Kiessling et al (2012) [which I was a coauthor].

All in all, I think that, although this paper tackles an important issue, it does not meet its potential because it is confusing. The focus on occurrences and ENM methods and not the questions posed is a big part of this confusion for me. Take for example, the LIG and modern diversity gradients. Although these are major parts of the the paper (goals 1 and 2), no where is diversity shown, only numbers of occurrences and numbers of habitable cells.

We actually show sample in bin diversity within Figure 1. The first aim refers to using ecological niche modelling to predict where we should or should not have fossil occurrences based on predicted habitability (not so much diversity). In regards to the second aim, we are simply using ecological niche modelling to show similarities between modern and LIG latitudinal habitability, and what this could suggest for the latitudinal biodiversity gradient during the LIG, and under future climatic conditions. Ecological niche modelling is not a tool for predicting biodiversity, but the distribution of habitable localities. Rationally, we can make some inferences on the impact of change in habitability for biodiversity, but not quantify how many of species x or y we should have.

Major comments:

I understand the motive for studying the relationships between coral distributions and climate. I don't think a convincing case for using niche models is made in the introduction. No doubt there are sampling issues with the past that cannot be fully ameliorated by sub-sampling. But framing the current paper as a methodological correction to Kiessling et al limits the current paper's appeal and impact. ENM are needed for a much more important reason than correcting Kiessling—a niche model is necessary for the forward prediction into future climate scenarios.

We don't feel this is a fair comment. This paper does not set out to correct Kiessling et al. (2012), but compliment it. For example, we concur that there was a higher diversity at higher latitudes during the LIG. However, the combination of predicted habitability at equatorial latitudes during the LIG and the known sampling bias within tropical latitudes, suggests the equatorial decline might be artificial. We have added a couple of sentences to the introduction (relating to the next point) to bolster the need for niche models as highlighted in this comment.

I suggest adding in a paragraph into the introduction, at about line 73, to point out that there is no other way to make a prediction into the future than with a well trained model. Adding in fossil occurrences from the last interglacial, when the earth was warmer than today, aids in the modeling effort because modern occurrences alone sample a small range of possible climatic variation. More sampled variation in temperatures and occurrences will lead to a higher fidelity model.

We have added “Future projections trained solely on modern-occurrences may underestimate habitability due a restricted range in possible climatic variation. Through incorporation of the fossil record this variation is increased, and might improve future projections by reducing the amount of novel climate conditions faced from a modern-only trained model.”

Geographic ranges surely are species specific, therefore I can't see how lumping occurrences of all coral species together can say anything about the range shifts from LIG to the modern. However, those ENMs are the only option for predicting the coral occurrences in the future. Would species specific ENMs be too noisy?

Our study is not species specific, but at ecotype-level as we specify in the methods. Running this type of global analysis at species-level would require higher resolution environmental data and fairer sampling between different species to achieve meaningful predictions based on a correlative approach. It is also worth mentioning that within different populations of the same species you may have different environmental tolerances due to historical experience (e.g. thermal regimes; Howells et al. (2013)), which ought to be considered if focusing efforts at species-level, at which point mechanistic models may prove more fruitful. In this paper we are not looking at specific geographic ranges sizes of species but the range size of suitable habitats.

I find understanding Figure 2 to be confusing. There are many combinations of training dataset and modeled pattern that are hard to tease apart. For one, I'm surprised that the modern-trained modern habitability is so different from the observed modern occurrences. The peak habitability in the southern hemisphere is 5 or more degrees off of the peak occurrences in the raw data. This mismatch makes me wonder if the number of habitable cells is informative for diversity. What is the pattern of latitudinal diversity in this dataset anyway?

Within this figure there are only two training datasets (a) modern-trained and (b) LIG-trained. The separate curves on each plot show different threshold selections. We have amended figure captions in the hope to make this clearer. These figures do not represent diversity in any way, and we do not refer to them in such a manner. We are simply looking at how latitudinal habitability changes under different climate scenarios, and what implications this may have for diversity. For example, if habitability is greater during the LIG at equatorial latitudes than today, it doesn't make sense that we would have an equatorial decline, at least due to abiotic reasons. In regards to the mismatch between habitability and occurrences, one should not expect a perfect match. To provide a few examples of why:

- You may find more occurrences in site A opposed to site B due to sampling
- Site A might have a higher level of connectivity than site B and receive more coral larvae
- Site B might be more environmentally suitable but is impacted by biotic and anthropogenic interactions
- Distribution is impacted by historical reasons such as limits to dispersal

We thank the reviewers for the kind and constructive comments they have provided to strengthen this manuscript. We feel that it has improved the quality of the manuscript considerably and look forward to any further comments.

Kind regards,

Lewis A. Jones,

On behalf of all the co-authors.

References

- Howells, E. J., Berkelmans, R., van Oppen, M. J. H., Willis, B. L., and Bay, L. K., 2013, Historical thermal regimes define limits to coral acclimatization: *Ecology*, v. 94, no. 5, p. 1078-1088.
- Kiessling, W., Simpson, C., Beck, B., Mewis, H., and Pandolfi, J. M., 2012, Equatorial decline of reef corals during the last Pleistocene interglacial: *Proceedings of the National Academy of Sciences*, v. 109, no. 52, p. 21378-21383.

[revised manuscript text omitted]

All supplementary material has been included as part of the associated Dryad package with this
manuscript: <https://datadryad.org/review?doi=doi:10.5061/dryad.2c0g95d>

**7 AUTHOR'S CONTRIBUTIONS**

LAJ and PDM conceived and designed the project; LAJ performed the analyses and interpretation
of data; PJV provided the Last Interglacial climate data; SK (Getech Group plc) provided the
global digital elevation model; LAJ, PDM, AF and PAA contributed to the writing of the
manuscript; LAJ produced the figures.

**8 COMPETING INTERESTS**

We declare we have no competing interests.

**9 FUNDING**

LAJ was supported by an Imperial College London President's PhD Scholarship. PDM's
contribution was supported by a Royal Society University Research Fellowship (UF160216).

**10 ACKNOWLEDGEMENTS**

We are grateful for the efforts of all those who have collected modern-day and Pleistocene reef
coral data, as well as those who have entered this data into the Ocean Biogeographic Information
System and the Paleobiology Database (in particular Wolfgang Kiessling). We would also like to
thank Getech Group plc for supplying the digital elevation model used in this study. Helpful
discussions with Neftalí Sillero (University of Porto), Elena Couce (Centre for Environment,
Fisheries and Aquaculture Science) and Alessandro Chiarenza (Imperial College London)

improved this work, as did comments from six anonymous reviewers. This is Paleobiology
Database official publication number XXX.

**11 REFERENCES**

- 1 IPCC. 2014 Climate Change 2014: Synthesis Report. Contribution of Working Groups I, II and
III to the Fifth Assessment Report of the Intergovernmental Panel on Climate Change. Geneva,
Switzerland: IPCC.
- 2 Poloczanska, E. S., Burrows, M. T., Brown, C. J., García Molinos, J., Halpern, B. S., Hoegh-
Guldberg, O., Kappel, C. V., Moore, P. J., Richardson, A. J., Schoeman, D. S., *et al.* 2016
Responses of Marine Organisms to Climate Change across Oceans. *Frontiers in Marine Science*.
**3**, (10.3389/fmars.2016.00062)
- 3 Poloczanska, E. S., Brown, C. J., Sydeman, W. J., Kiessling, W., Schoeman, D. S., Moore, P.
478 J., Brander, K., Bruno, J. F., Buckley, L. B., Burrows, M. T., *et al.* 2013 Global imprint of
479 climate change on marine life. *Nature Climate Change*. **3**, 919-925. (10.1038/nclimate1958)
- García Molinos, J., Halpern, Benjamin S., Schoeman, David S., Brown, Christopher J.,
Kiessling, W., Moore, Pippa J., Pandolfi, John M., Poloczanska, Elvira S., Richardson,
Anthony J., Burrows, Michael T. 2015 Climate velocity and the future global redistribution of
marine biodiversity. *Nature Climate Change*. **6**, 83-88. (10.1038/nclimate2769)
- Heron, S. F., Maynard, J. A., van Hooidonk, R., Eakin, C. M. 2016 Warming Trends and
Bleaching Stress of the World's Coral Reefs 1985-2012. *Sci Rep*. **6**, 38402. (10.1038/srep38402)
- Hughes, T. P., Kerry, J. T., Alvarez-Noriega, M., Alvarez-Romero, J. G., Anderson, K. D.,
Baird, A. H., Babcock, R. C., Bejer, M., Bellwood, D. R., Berkelmans, R., *et al.* 2017 Global
warming and recurrent mass bleaching of corals. *Nature*. **543**, 373-377. (10.1038/nature21707)
- Pandolfi, J. M., Bradbury, R. H., Sala, E., Hughes, T. P., Bjorndal, K. A., Cooke, R. G.,
McArdle, D., McClenachan, L., Newman, M. J. H., Paredes, G., *et al.* 2003 Global Trajectories
of the Long-Term Decline of Coral Reef Ecosystems. *Science*. **301**, 955-958.
- Grupstra, C. G. B., Coma, R., Ribes, M., Leydet, K. P., Parkinson, J. E., McDonald, K., Catllà,
493 M., Voolstra, C. R., Hellberg, M. E., Coffroth, M. A. 2017 Evidence for coral range expansion
accompanied by reduced diversity of Symbiodinium genotypes. *Coral Reefs*. **36**, 981-985.
(10.1007/s00338-017-1589-2)
- Yamano, H., Sugihara, K., Nomura, K. 2011 Rapid poleward range expansion of tropical reef
corals in response to rising sea surface temperatures. *Geophysical Research Letters*. **38**, 1-6.
(10.1029/2010gl046474)
- Muir, P. R., Wallace, C. C., Done, T., Aguirre, J. D. 2015 Limited scope for latitudinal
extension of reef corals. *Science*. **348**,
- Baird, A. H., Sommer, B., Madin, J. S. 2012 Pole-ward range expansion of *Acropora* spp.
along the east coast of Australia. *Coral Reefs*. **31**, 1063-1063. (10.1007/s00338-012-0928-6)
- Precht, W. F., Aronson, R. B. 2004 Climate flickers and range shifts of reef corals. *Frontiers*
*in Ecology and the Environment*. **2**, 307-314.
- Kiessling, W., Simpson, C., Beck, B., Mewis, H., Pandolfi, J. M. 2012 Equatorial decline of
reef corals during the last Pleistocene interglacial. *Proceedings of the National Academy of*
*Sciences*. **109**, 21378-21383.
- Couce, E., Ridgwell, A., Hendy, E. J. 2012 Environmental controls on the global distribution
of shallow-water coral reefs. *Journal of Biogeography*. **39**, 1508-1523. (10.1111/j.1365-
2699.2012.02706.x)

Couce, E., Ridgwell, A., Hendy, E. J. 2013 Future habitat suitability for coral reef ecosystems
under global warming and ocean acidification. *Glob Chang Biol.* **19**, 3592-3606.
(10.1111/gcb.12335)

Freeman, L. A., Kleypas, J. A., Miller, A. I. 2013 Coral Reef Habitat Response to Climate
Change Scenarios. *PLOS ONE.* **8**, 1-14. (10.1371/)

Freeman, L. A. 2015 Robust Performance of Marginal Pacific Coral Reef Habitats in Future
Climate Scenarios. *PLoS One.* **10**, e0128875. (10.1371/journal.pone.0128875)

Dietl, G. P., Kidwell, S. M., Brenner, M., Burney, D. A., Flessa, K. W., Jackson, S. T., Koch,
P. L. 2015 Conservation Paleobiology: Leveraging Knowledge of the Past to Inform
Conservation and Restoration. *Annual Review of Earth and Planetary Sciences.* **43**, 79-103.
(10.1146/annurev-earth-040610-133349)

Hutchinson, G. E. 1957 Concluding Remarks. *Cold Spring Harbor Symposia Quantitative*
*Biology.* **22**, 415-427.

Lima-Ribeiro, M. S., Moreno, A. K. M., Terribile, L. C., Caten, C. T., Loyola, R., Rangel, T.
F., Diniz-Filho, J. A. F., VanDerWal, J. 2017 Fossil record improves biodiversity risk assessment
under future climate change scenarios. *Diversity and Distributions.* (10.1111/ddi.12575)

Maguire, K. C., Nieto-Lugilde, D., Blois, J. L., Fitzpatrick, M. C., Williams, J. W., Ferrier, S.,
Lorenz, D. J. 2016 Controlled comparison of species- and community-level models across novel
climates and communities. *Proc Biol Sci.* **283**, 20152817. (10.1098/rspb.2015.2817)

Sofaer, H. R., Jarnevich, C. S., Flather, C. H. 2018 Misleading prioritizations from modelling
range shifts under climate change. *Global Ecology and Biogeography.* (10.1111/geb.12726)

Hortal, J., Jiménez-Valverde, A., Gómez, J., Lobo, J., Baselga, A. 2008 Historical bias in
biodiversity inventories affects the observed environmental niche of the species. *Oikos.* **117**,
847-858. (10.1111/j.2008.0030-1299.16434.x)

Greenstein, B. J., Pandolfi, J. M. 2008 Escaping the heat: range shifts of reef coral taxa in
coastal Western Australia. *Global Change Biology.* **14**, 513-528. (10.1111/j.1365-
2486.2007.01506.x)

Muhs, D. R., Simmons, K. R., Steinke, B. 2002 Timing and warmth of the Last Interglacial
period: new U-series evidence from Hawaii and Bermuda and a new fossil compilation for North
America. *Quaternary Science Reviews.* **21**, 1355-1383.

Clark, P. U., Huybers, P. 2009 Interglacial and future sea level. *Nature.* **462**, 856-857.
(10.1029/2009gl040222)

McKay, N. P., Overpeck, J. T., Otto-Bliesner, B. L. 2011 The role of ocean thermal
expansion in Last Interglacial sea level rise. *Geophysical Research Letters.* **38**, n/a-n/a.
(10.1029/2011gl048280)

Hoffman, J. S., Clark, P. U., Parnell, A. C., He, F. 2017 Regional and global sea-surface
temperatures during the last interglaciation. *Science.* **355**, 276-279.

Alroy, J. 2010 Geographical, environmental and intrinsic biotic controls on Phanerozoic
marine diversification. *Palaeontology.* **53**, 1211-1235. (10.1111/j.1475-4983.2010.01011.x)

Saupe, E. E., Hendricks, J. R., Portell, R. W., Dowsett, H. J., Haywood, A., Hunter, S. J.,
Lieberman, B. S. 2014 Macroevolutionary consequences of profound climate change on niche
evolution in marine molluscs over the past three million years. *Proc Biol Sci.* **281**,
(10.1098/rspb.2014.1995)

Guinan, J., Brown, C., Dolan, M. F. J., Grehan, A. J. 2009 Ecological niche modelling of the
distribution of cold-water coral habitat using underwater remote sensing data. *Ecological*
*Informatics.* **4**, 83-92. (10.1016/j.ecoinf.2009.01.004)

Soto-Centeno, J. A., Steadman, D. W. 2015 Fossils reject climate change as the cause of
extinction of Caribbean bats. *Sci Rep.* **5**, 7971. (10.1038/srep07971)

Davis, E. B., McGuire, J. L., Orcutt, J. D. 2014 Ecological niche models of mammalian
glacial refugia show consistent bias. *Ecography*. n/a-n/a. (10.1111/ecog.01294)

Kleypas, J. A., McManus, J. W., Meñez, L. A. B. 1999 Environmental Limits to Coral Reef
Development: Where Do We Draw the Line? *American Zoologist*. **39**, 146-159.

Veron, J. E. N. 2000 *Corals of the World*. Townsville, Australia: Australian Institute of
Marine Science.

Valdes, P. J., Armstrong, E., Badger, M. P. S., Bradshaw, C. D., Bragg, F., Davies-Barnard,
566 T., Day, J. J., Farnsworth, A., Hopcroft, P. O., Kennedy, A. T., *et al.* 2017 The BRIDGE
HadCM3 family of climate models: HadCM3@Bristol v1.0. *Geoscientific Model Development*
*Discussions*. 1-42. (10.5194/gmd-2017-16)

IPCC. 2013 Climate Change 2013: The Physical Science Basis. Contribution of Working
Group I to the Fifth Assessment Report of the Intergovernmental Panel on Climate Change.
Cambridge University Press, Cambridge, United Kingdom and New York, NY, USA, .

Sheppard, C. R. 2003 Predicted recurrences of mass coral mortality in the Indian Ocean.
*Nature*. **425**, 294-297.

Singarayer, J. S., Valdes, P. J. 2010 High-latitude climate sensitivity to ice-sheet forcing over
the last 120kyr. *Quaternary Science Reviews*. **29**, 43-55. (10.1016/j.quascirev.2009.10.011)

Stone, E. J., Lunt, D. J., Annan, J. D., Hargreaves, J. C. 2013 Quantification of the Greenland
ice sheet contribution to Last Interglacial sea level rise. *Climate of the Past*. **9**, 621-639.
(10.5194/cp-9-621-2013)

Capron, E., Govin, A., Stone, E. J., Masson-Delmotte, V., Mulitza, S., Otto-Bliesner, B.,
Rasmussen, T. L., Sime, L. C., Waelbroeck, C., Wolff, E. W. 2014 Temporal and spatial
structure of multi-millennial temperature changes at high latitudes during the Last Interglacial.
*Quaternary Science Reviews*. **103**, 116-133. (10.1016/j.quascirev.2014.08.018)

Holloway, M. D., Sime, L. C., Singarayer, J. S., Tindall, J. C., Valdes, P. J. 2016
Reconstructing paleosalinity from $\delta^{18}O$: Coupled model simulations of the Last Glacial
Maximum, Last Interglacial and Late Holocene. *Quaternary Science Reviews*. **131**, 350-364.
(10.1016/j.quascirev.2015.07.007)

van Vuuren, D. P., Edmonds, J., Kainuma, M., Riahi, K., Thomson, A., Hibbard, K., Hurtt, G.
C., Kram, T., Krey, V., Lamarque, J.-F., *et al.* 2011 The representative concentration pathways:
an overview. *Climatic Change*. **109**, 5-31. (10.1007/s10584-011-0148-z)

Kearney, M. R., Wintle, B. A., Porter, W. P. 2010 Correlative and mechanistic models of
species distribution provide congruent forecasts under climate change. *Conservation Letters*. **3**,
203-213. (10.1111/j.1755-263X.2010.00097.x)

Warren, D. L., Glor, R. E., Turelli, M. 2008 Environmental niche equivalency versus
conservatism: quantitative approaches to niche evolution. *Evolution*. **62**, 2868-2883.
(10.1111/j.1558-5646.2008.00482.x)

Broennimann, O., Fitzpatrick, M. C., Pearman, P. B., Petitpierre, B., Pellissier, L., Yoccoz, N.
G., Thuiller, W., Fortin, M.-J., Randin, C., Zimmermann, N. E., *et al.* 2012 Measuring ecological
niche overlap from occurrence and spatial environmental data. *Global Ecology and*
*Biogeography*. **21**, 481-497. (10.1111/j.1466-8238.2011.00698.x)

Schoener, T. W. 1968 The Anolis Lizards of Bimini: Resource Partitioning in a Complex
Fauna. *Ecology*. **49**, 704-726. (10.2307/1935534)

Di Cola, V., Broennimann, O., Petitpierre, B., Breiner, F. T., D'Amen, M., Randin, C., Engler,
R., Pottier, J., Pio, D., Dubuis, A., *et al.* 2017 ecospat: an R package to support spatial analyses
and modeling of species niches and distributions. *Ecography*. **40**, 774-787.
(10.1111/ecog.02671)

Booth, T. H., Nix, H. A., Busby, J. R., Hutchinson, M. F., Franklin, J. 2014 bioclim: the first
species distribution modelling package, its early applications and relevance to most
currentMaxEntstudies. *Diversity and Distributions*. **20**, 1-9. (10.1111/ddi.12144)

Liaw, A., Wiener, M. 2002 Classification and Regression by randomForest. *R News*. **2**, 18-22.

Phillips, S. J., Anderson, R. P., Dudík, M., Schapire, R. E., Blair, M. E. 2017 Opening the
black box: an open-source release of Maxent. *Ecography*. (10.1111/ecog.03049)

Waterson, A. M., Schmidt, D. N., Valdes, P. J., Holroyd, P. A., Nicholson, D. B., Farnsworth,
613 A., Barrett, P. M. 2016 Modelling the climatic niche of turtles: a deep-time perspective. *Proc*
*Biol Sci*. **283**, (10.1098/rspb.2016.1408)

Chiarenza, A. A., Mannion, P. D., Lunt, D. J., Farnsworth, A., Jones, L. A., Kelland, S. J.,
Allison, P. A. 2019 Ecological niche modelling does not support climatically-driven dinosaur
diversity decline before the Cretaceous/Paleogene mass extinction. *Nat Commun*. **10**, 1091.
(10.1038/s41467-019-08997-2)

Thuiller, W., Lafourcade, B., Engler, R., Araújo, M. B. 2009 BIOMOD - a platform for
ensemble forecasting of species distributions. *Ecography*. **32**, 369-373. (10.1111/j.1600-
0587.2008.05742.x)

Thuiller, W., Georges, D., Engler, R., Breiner, F. biomod2: Ensemble Platform for Species
Distribution Modeling. R package version 3.3-7 ed 2016.

Peterson, A. T., Papeş, M., Soberón, J. 2008 Rethinking receiver operating characteristic
analysis applications in ecological niche modeling. *Ecological Modelling*. **213**, 63-72.
(10.1016/j.ecolmodel.2007.11.008)

Allouche, O., Tsoar, A., Kadmon, R. 2006 Assessing the accuracy of species distribution
models: prevalence, kappa and the true skill statistic (TSS). *Journal of Applied Ecology*. **43**,
1223-1232. (10.1111/j.1365-2664.2006.01214.x)

Swets, J. A. 1988 Measuring the Accuracy of Diagnostic Systems. *Science*. **240**, 1285-1293.

Williams, W. J., Jackson, S. T. 2007 Novel climates, no-analog communities, and ecological
surprises. *Frontiers in Ecology and the Environment*. **5**, 475-482. (10.1890/070037)

Jeffers, D., Willis, K. J. 2014 Vegetation response to climate change during the Last
Interglacial–Last Glacial transition in the southern Bekaa Valley, Lebanon. *Palynology*. **38**, 195-
206. (10.1080/01916122.2014.880958)

Tyrberg, T. 2010 Avifaunal responses to warm climate: the message from Last Interglacial
faunas. In Proceedings of the VII International Meeting of the Society of Avian Paleontology
and Evolution, ed. W.E. Boles and T.H. Worthy. *Records of the Australian Museum*. **62**, 193-
205. (10.3853/j.0067-1975.62.2010.1543)

Veron, J. E. N., Devantier, L. M., Turak, E., Green, A. L., Kininmonth, S., Stafford-Smith,
641 M., Peterson, N. 2009 Delineating the Coral Triangle. *Galaxea, Journal of Coral Reef Studies*.
**11**, 91-100.

Spalding, M. D., Ravilious, C., Green, E. P. 2001 *World atlas of coral reefs. Prepared at the*
*UNEP world conservation monitoring centre*. Berkeley, CA, USA: University of California
Press.

Hoegh-Guldberg, O. 2010 Coral reef ecosystems and anthropogenic climate change. *Regional*
*Environmental Change*. **11**, 215-227. (10.1007/s10113-010-0189-2)

Crame, J. A. 2001 Taxonomic Diversity Gradients through Geological Time. *Diversity and*
*Distributions*. **7**, 175-189.

Mannion, P. D., Upchurch, P., Benson, R. B., Goswami, A. 2014 The latitudinal biodiversity
gradient through deep time. *Trends Ecol Evol*. **29**, 42-50. (10.1016/j.tree.2013.09.012)

Vilhena, D. A., Smith, A. B. 2013 Spatial bias in the marine fossil record. *PLoS One*. **8**,
e74470. (10.1371/journal.pone.0074470)

Kiessling, W. 2005 Habitat effects and sampling bias on Phanerozoic reef distribution.
*Facies*. **51**, 24-32. (10.1007/s10347-004-0044-3)

Yasuhara, M., Hunt, G., Dowsett, H. J., Robinson, M. M., Stoll, D. K. 2012 Latitudinal
species diversity gradient of marine zooplankton for the last three million years. *Ecol Lett*. **15**,
1174-1179. (10.1111/j.1461-0248.2012.01828.x)

McCulloch, M. T., Esat, T. 2000 The coral record of last interglacial sea levels and sea
surface temperatures. *Chemical Geology*. **169**, 107-129.

Dalbeck, P., Cusack, M., Dobson, P. S., Allison, N., Fallick, A. E., Tudhope, A. W. 2011
Identification and composition of secondary meniscus calcite in fossil coral and the effect on
predicted sea surface temperature. *Chemical Geology*. **280**, 314-322.
(10.1016/j.chemgeo.2010.11.018)

Brocas, W. M., Felis, T., Obert, J. C., Gierz, P., Lohmann, G., Scholz, D., Kölling, M.,
Scheffers, S. R. 2016 Last interglacial temperature seasonality reconstructed from tropical
Atlantic corals. *Earth and Planetary Science Letters*. **449**, 418-429.
(10.1016/j.epsl.2016.06.005)

Brocas, W. M., Felis, T., Gierz, P., Lohmann, G., Werner, M., Obert, J. C., Scholz, D.,
Kölling, M., Scheffers, S. R. 2018 Last Interglacial Hydroclimate Seasonality Reconstructed
From Tropical Atlantic Corals. *Paleoceanography and Paleoclimatology*. **33**, 198-213.
(10.1002/2017pa003216)

Warren, R., VanDerWal, J., Price, J., Welbergen, J. A., Atkinson, I., Ramirez-Villegas, J.,
Osborn, T. J., Jarvis, A., Shoo, L. P., Williams, S. E., *et al.* 2013 Quantifying the benefit of early
climate change mitigation in avoiding biodiversity loss. *Nature Climate Change*. **3**, 678-682.
(10.1038/nclimate1887)

Sunday, J. M., Bates, A. E., Dulvy, N. K. 2012 Thermal tolerance and the global
redistribution of animals. *Nature Climate Change*. **2**, 686-690. (10.1038/nclimate1539)

Jablonski, D., Belanger, C. L., Berke, S. K., Huang, S., Krug, A. Z., Roy, K., Tomasovych,
680 A., Valentine, J. W. 2013 Out of the tropics, but how? Fossils, bridge species, and thermal ranges
in the dynamics of the marine latitudinal diversity gradient. *Proc Natl Acad Sci U S A*. **110**,
10487-10494. (10.1073/pnas.1308997110)

van Hooijdonk, R., Maynard, J. A., Manzello, D., Planes, S. 2014 Opposite latitudinal
gradients in projected ocean acidification and bleaching impacts on coral reefs. *Glob Chang Biol*.
**20**, 103-112. (10.1111/gcb.12394)

Jiang, L.-Q., Feely, R. A., Carter, B. R., Greeley, D. J., Gledhill, D. K., Arzayus, K. M. 2015
Climatological distribution of aragonite saturation state in the global oceans. *Global*
*Biogeochemical Cycles*. **29**, 1656-1673. (10.1002/2015gb005198)

Peterson, A. T., Soberón, J., Pearson, R. G., Anderson, R. P., Martínez, -. M., E., Nakamura,
690 M., Araújo, M. B. 2011 *Ecological Niches and Geographic Distributions*. Oxford: Princeton
University Press.

Soberón, J., Nakamura, M. 2009 Niches and distributional areas: concepts, methods, and
assumptions. *Proc Natl Acad Sci U S A*. **106 Suppl 2**, 19644-19650.
(10.1073/pnas.0901637106)

**12 FIGURE CAPTIONS**

Figure 1. Latitudinal and global distribution of modern and LIG reef coral occurrences. (a-b) Frequency of
occurrence records (Occ), species diversity (SIB) and shared species (species present in both time slices
[sSIB]) of reef corals binned at 5° latitudinal zones. (c) World distribution map of shared species
(subsampled at 1.25° x 1.25° and clipped to environmental data) between the modern and the LIG. Grey
shaded area represents the extent of the tropics in the present day. Dashed lines indicate the equator, the
Northern Tropic, and the Southern Tropic.

Figure 2. Latitudinal plots for predicted habitability from ecological niche modelling, using three binary
thresholds: the 5th-, 10th-, and 20th-percentiles of the lowest tail of habitat suitability values (5LPT, 10LPT,
and 20LPT). Plots indicate habitability for the Last Interglacial (LIG), modern, representative concentration
pathway 4.5 (RCP4.5), and 8.5 (RCP8.5). In this instance, habitability is the sum of cells designated as
habitable, within 5° latitudinal bins, by the mean prediction, trained on the (a) modern occurrence dataset,
and (b) LIG occurrence dataset. Grey shaded area represents the extent of the tropics in the present day.
Dashed lines indicating the equator, the Northern Tropic, and the Southern Tropic.

Figure 3. Combined binary maps (using threshold 20LPT) showing the potential distribution of reef corals
under LIG, modern, RCP4.5 and RCP8.5 climate regimes. (a) Global maps of habitability under the four
climate scenarios. Grey cells represent the potential distribution predicted through using only the modern
reef coral training dataset. Yellow cells characterise cells gained when coupling LIG and modern trained
predictions. Blue cells indicate unsuitable locations, whilst black indicates no data cells (landmass). (b)
Habitability maps clipped to Indo-Pacific for all climate scenarios to highlight change in habitability within
the coral triangle. Yellow cells highlight suitable habitats, whilst blue cells indicate unsuitable locations,
and black indicates no data cells (landmass).